# UNBIASED LEARNING WITH STATE-CONDITIONED REWARDS IN ADVERSARIAL IMITATION LEARNING

## ABSTRACT

Adversarial imitation learning has emerged as a general and scalable framework for automatic reward acquisition. However, we point out that previous methods commonly exploited occupancy-dependent reward learning formulation—which hinders the reconstruction of optimal decision as an energy-based model. Despite the theoretical justification, the occupancy measures tend to cause issues in practice because of high variance and low vulnerability to domain shifts. Another reported problem is termination biases induced by provided rewarding and regularization schemes around terminal states. In order to deal with these issues, this work presents a novel algorithm called causal adversarial inverse reinforcement learning. Our formulation draws a strong connection between adversarial learning and energy-based reinforcement learning; thus, the architecture is capable of recovering a reward function that induces a multi-modal policy. In experiments, we demonstrate that our approach outperforms prior methods in challenging continuous control tasks, even under significant variation in the environments.

## 1 INTRODUCTION

Inverse reinforcement learning (IRL) is an algorithm of recovering the ground truth reward function from observed behavior (Ng & Russell, 2000). IRL algorithms—followed by appropriate reinforcement learning (RL) algorithms—can optimize policy through farsighted cumulative value measures in the given system (Sutton & Barto, 2018); hence it can usually achieve more satisfying results than mere supervision. While a few studies have investigated recovering reward functions to continuous spaces (Babes et al., 2011; Levine & Koltun, 2012), IRL algorithms often fail to find the ground-truth reward function in high-dimensional complex domains (Finn et al., 2016b).

The notion of the ground-truth reward requires elaboration since IRL is an ill-posed problem; there can be numerous solutions to the reward function inducing the same optimal policy (Ng et al., 1999; Ng & Russell, 2000). Recently, adversarial imitation learning (AIL) as a reward acquisition method has shown promising results (Ho & Ermon, 2016). One of the distinctive strengths of AIL is the scalability through parameterized non-linear functions such as neural networks.

The maximum causal entropy principles are widely regarded as the solution when the optimal control problem is modeled as probabilistic inference (Ziebart et al., 2010; Haarnoja et al., 2017). In particular, probabilistic modeling using a continuous energy function forms a representation called an energy-based model (EBM). We highlight the following advantages of the energy-based IRL:

- It provides a unified framework for stochastic policies to the learning; most probabilistic models can be viewed as special types of EBMs (LeCun et al., 2006).
- It rationalizes the stochasticity of behavior; this provides robustness in the face of uncertain dynamics (Ziebart et al., 2010) and a natural way of modeling complex multi-modal distribution.

AIL reward functions seem to be exceptions to these arguments—the AIL framework produces distinct types of rewards that are ever-changing and are intended for discriminating joint densities. We argue that these characteristics hinder proper information projection to the optimal decision.

This work points out that there remain two kinds of biases in AIL. The established AIL algorithms are typically formalized by the cumulative densities called *occupancy measure*. We claim that the accumulated measures contain biases that are not related to modeling purposeful behavior, and the formulation is vulnerable to distributional shifts of an MDP. Empirically, they work as dominant

noises in training because of the formulation's innate high variance. The other bias is implicit survival or early termination bias caused by reward formulation, which lacks consideration for the terminal states in finite episodes. These unnormalized rewards often provokes sub-optimal behaviors where the agent learns to maliciously make use of temporal-aware strategies.

This paper proposes an adversarial IRL method called causal adversarial inverse reinforcement learning (CAIRL). We primarily associate the reward acquisition method with approaches for energy-based RL and IRL algorithms; the CAIRL reward function can induce complex probabilistic behaviors with multiple modalities. We then show that learning with a dual discriminator architecture provides stepwise, state-conditioned rewards. For handling biases induced by finite-horizon, the model postulates the reward function satisfies a Bellman equation, including "self-looping" terminal states. As a result, it learns the reward function satisfying the property of EBMs.

Noteworthy contributions of this work are 1) a model-free, energy-based IRL algorithm that is effective in high-dimensional environments, 2) a dual discriminator architecture for recovering a robust state-conditioned reward function, 3) an effective approach for handling terminal states, and 4) meaningful experiments and comparison studies with state-of-the-art algorithms in various topics.

## 2 RELATED WORKS

Imitation learning is a fundamental approach for modeling intellectual behavior from an expert at specific tasks (Pomerleau, 1991; Zhang et al., 2018). For the standard framework called Behavioral Cloning, learning from demonstrations is treated as supervised learning for a trajectory dataset. On the other hand, IRL aims to study the reward function of the underlying system, which characterizes the expert. In this perspective, training a policy with an IRL reward function is a branch of imitation learning, specialized in dealing with sequential decision-making problems by recovering the concise representation of a task (Ng & Russell, 2000; Abbeel & Ng, 2004).

For modeling stochastic expert policies, Boltzmann distributions appeared in early IRL research, such as Bayesian IRL, natural gradient IRL, and maximum likelihood IRL (Ramachandran & Amir, 2007; Neu & Szepesvári, 2012; Babes et al., 2011). Notably, maximum entropy IRL (Ziebart et al., 2008) is explicitly formulated based on the principle of maximum entropy. The framework has also been derived from causal entropy—the derived algorithm can model the purposeful distribution of optimal policy into a reward function (Ziebart et al., 2010). Our work draws significant inspirations from these prior works and aims to redeem the perspective of probabilistic causality.

Recently, AIL methods (Ho & Ermon, 2016; Fu et al., 2017; Ghasemipour et al., 2020) have shown great success on continuous control benchmarks. Each of the them provides a unique divergence minimization scheme by its architecture. In particular, our work shares major components with AIRL. It has been argued that the algorithm does not recover the energy of the expert policy (Liu et al., 2020). We stress that our work introduces essential concepts to draw an energy-based representation of the expert policy correctly. The discriminator design is based on the rich energy-based interpretation of GANs (Zhao et al., 2016; Azadi et al., 2018; Che et al., 2020) and numerous studies with multiple discriminators (Chongxuan et al., 2017; Gan et al., 2017; Choi et al., 2018).

The issues of finite-horizon tasks were initially raised in RL during the discussion of time limits in MDP benchmarks (Pardo et al., 2017; Tucker et al., 2018). It turned out that the time limits, or even the existence of terminal states, would significantly affect the value learning procedure of RL compared to that generated in infinite horizon MDPs. IRL suffers from the identical problem that reward learning of finite episodes is not really stable for tasks outside of appropriate benchmarks. Kostrikov et al. (2018) suggested explicitly adding a self-repeating *absorbing* state (Sutton & Barto, 2018) after the terminal state; consequently, AIL discriminators can evaluate the termination frequencies.

## 3 BACKGROUND

**Markov Decision Process (MDP)**. We define an MDP as a tuple $M = (\mathcal{S}, \mathcal{A}, \mathcal{P}, r, p_0, \gamma)$ where $\mathcal{S}$ and $\mathcal{A}$ denote the state and action spaces, and $\gamma$ is the discount factor. The transition distribution $\mathcal{P}$, the deterministic state-action reward function $r$, and the initial state distribution $p_0$ are unknown. Let $\tau_\pi$ and $\tau_E$ be sequences of finite states and actions $(s_0, a_0, \ldots, a_{T-1}, s_T)$ obtained by a policy $\pi$ and the expert policy $\pi_E$, respectively. The term $\rho_\pi$ denotes the occupancy measures derived by

Table 1: The objectives for AIL algorithms in a form as the minimization of statistical divergences.

| Method | Optimized Objective (Minimization) |
|---|---|
| Behavioral Cloning | $\mathbb{E}_{\pi_E}\big[D_{\mathrm{KL}}\big(\pi_E(a\vert s)\Vert\pi(a\vert s)\big)\big] = -\mathbb{E}_{\pi_E}[\log\pi(a\vert s)] + \mathrm{const}$ |
| GAIL (Ho & Ermon, 2016) | $\mathbb{E}_{\pi}\big[D_{\mathrm{JS}}\big(\rho_\pi(s,a),\rho_E(s,a)\big) - \mathcal{H}(\pi(\cdot\vert s))\big]$ |
| AIRL (Fu et al., 2017) | $\mathbb{E}_{\pi}\big[D_{\mathrm{KL}}\big(\rho_\pi(s,a)\Vert\rho_E(s,a)\big)\big] = -\mathbb{E}_{\pi}\big[\log\rho_E(s,a) + \mathcal{H}(\rho_\pi)\big]$ |
| FAIRL (Ghasemipour et al., 2020) | $\mathbb{E}_{\pi}\big[D_{\mathrm{KL}}\big(\rho_E(s,a)\Vert\rho_\pi(s,a)\big)\big] = -\mathbb{E}_{\pi_E}\big[\log\rho_\pi(s,a) + \mathcal{H}(\rho_E)\big]$ |
| CAIRL (this work) | $\mathbb{E}_{\pi}\big[D_{\mathrm{KL}}\big(\pi(a\vert s)\Vert\pi_E(a\vert s)\big)\big] = -\mathbb{E}_{\pi}[\mathbf{r}(s,a) + \mathcal{H}(\pi(\cdot\vert s))] + \mathrm{const}$ |

$\pi$, and is defined as $\rho_\pi(s,a) = \pi(a\vert s)\sum_{t=0}^{\infty}\gamma^t\Pr(s_t = s\vert\pi)$. With a slight abuse of notation, we refer to the occupancy measures of states as $\rho_E(s)$ and $\rho_\pi(s)$. The expectation of $\pi$ for an arbitrary function $c$ denotes an expected return for infinite-horizon: $\mathbb{E}_\pi[c(s,a)] \triangleq \mathbb{E}[\sum_{t=0}^{\infty}\gamma^t c(s,a)\vert\pi]$.

**Maximum Entropy IRL (MaxEnt IRL)**. Ziebart (2010) and Haarnoja et al. (2017) defined the optimality of stochastic policy with an entropy-regularized RL objective as follows:

$$\pi^\star = \arg\max_{\pi\in\Pi}\sum_t \mathbb{E}_{(s_t,a_t)\sim\rho_\pi}\big[r(s_t,a_t) + \alpha\mathcal{H}(a_t\vert s_t)\big]$$

where $\mathcal{H}$ denotes the causal entropy function.[1] If $\pi_E$ is the MaxEnt RL policy, the softmax Bellman optimality equations can be defined by the following recursive logic:

$$\begin{aligned}Q^\star(s_t,a_t) &= r(s_t,a_t) + \gamma\mathbb{E}_{s_{t+1}\sim\mathcal{P}(\cdot\vert s_t,a_t)}\big[V^\star(s_{t+1})\big]\\ V^\star(s_t) &= \mathbb{E}_{a_t\sim\pi_E(\cdot\vert s_t)}\big[Q^\star(s_t,a_t) - \log\pi_E(a_t\vert s_t)\big]\end{aligned} \quad (1)$$

MaxEnt IRL algorithms (Ziebart et al., 2008; 2010) are energy-based interpretations of IRL which aim to find behavior abiding the MaxEnt principle. Such algorithms, however, are difficult to be computed when the given spaces are continuous or dynamics are unknown (Finn et al., 2016a).

**Adversarial Imitation Learning**. Ho & Ermon (2016) considered adversarial learning as a model-free, sampling-based approximation to MaxEnt IRL. Instead of exhaustively solving the problem, GAIL performs imitation learning by minimizing the divergence between the state-action occupancy measures from expert and learner through the following logistic objective:

$$\min_{\pi\in\Pi}\max_{D}\mathbb{E}_{\pi_E}\big[\log D(s,a)\big] + \mathbb{E}_{\pi}\big[\log\big(1 - D(s,a)\big)\big] - \mathcal{H}(\pi) \quad (2)$$

where $D \in (0,1)^{\vert\mathcal{S}\Vert\mathcal{A}\vert}$ indicates a binary classifier trained to distinguish between $\tau_\pi$ and $\tau_E$. The AIRL discriminator tries to disentangle a reward function that is invariant to dynamics. It takes a particular form: $D_\theta(s,a) = \exp(f_{\theta,\psi}(s,a))/(\exp(f_{\theta,\psi}(s,a)) + \pi_\phi(a\vert s))$. Learning with the AIRL can be considered as the reverse KL divergence between occupancy measures. Ghasemipour et al. (2020) proposed the FAIRL algorithm as an adversarial method for the forward KL divergence.

## 4 Unbiased Reinforcement Signals for Energy-Based Models

Our aim in this section is to investigate unbiased probabilistic modeling of the causality of decisions using the MaxEnt framework. In Sec. 4.1, we discuss the energy-based reward function. Sec. 4.2 and Sec. 4.3 introduce the modeling method of the particular reward function.

### 4.1 Energy-based Reward Representation

To accurately manage the MaxEnt framework, the EBM of expert is set to $\pi_E(a_t\vert s_t) \propto \exp\{-E(s_t,a_t)\} = \exp\{Q(s_t,a_t)\}$, such that the likelihood of distribution is proportional to the soft Q-function. The MaxEnt RL processes can be interpreted as minimizing the expected KL-divergence using the information projection (Haarnoja et al., 2017; 2018):

$$J(\pi) = \mathbb{E}_{s\sim\rho_\pi}\Big[D_{\mathrm{KL}}\Big(\pi(\cdot\vert s)\Big\Vert\tfrac{\exp\{Q(s,\cdot)\}}{Z(s)}\Big)\Big].$$

---

[1]For the rest of the paper, we occasionally omit the parameter with $\alpha = 1$ for simplicity of derivation.

Apparently in IRL, since the ground-truth reference will be the expert policy, the general objective of imitation learning: $\pi^\star = \arg\min_\pi \mathbb{E}_\pi[\mathrm{D}_{\mathrm{KL}}(\pi(\cdot|s)\|\pi_E(\cdot|s))]$, is indeed the special type of the MaxEnt RL objectives. We describe a *state-conditioned*, energy-based reward as a representation:

- If $\gamma \approx 0$, it formulates the "myopic" 1-step conditional KL divergence: $\mathrm{D}_{\mathrm{KL}}[\pi(\cdot|s)\|\pi_E(\cdot|s)]$.
- If $\gamma \approx 1$, assuming the dynamics are identical, it leads to the "far-sighted" cumulative densities:

$\mathbb{E}[\sum_{t=0}^\infty \mathrm{D}_{\mathrm{KL}}[\pi(\cdot|s_t)\|\pi_E(\cdot|s_t)]|\pi] = \mathrm{D}_{\mathrm{KL}}\big[\Pr(a_0, s_1, \dots|s_0 = s, \pi)\|\Pr(a_0, s_1, \dots|s_0 = s, \pi_E)\big]$.

By the discount factor $\gamma$, we can control how much subsequent steps we want to model. Therefore, the energy-based rewards generalize learning probabilistic inference of conditional decisions.

Note that AIL reward functions, curated in Table 1, usually do not retain these properties. For example, the AIRL reward function, namely: $f(s, a) = \log(\rho_E(s, a)/\rho_\pi(s))$, recovers the expert policy, only in the trivial case $\gamma = 0$, as $\pi_E(a|s) = \frac{f(s,a)}{\sum_{a'} f(s,a')}$. For $\gamma > 0$, the projection generally is not $\pi_E$, and also it is difficult to be analyzed. We also highlight that the standard MaxEnt RL with AIL rewards may not precisely recover the EBM. The more discussions are addressed in Appendix B.2.

## 4.2 State-Conditioned Rewards on the Principle of Maximum Entropy

We clarify a valid candidate set of reward functions when entropy-regularization is applied. One trivial solution of such function is the log-likelihood of expert $\mathbf{r}^\star(s, a) = \log \pi_E(a|s)$, in the condition: $V^\star(s) = V^\star(s') = 0$. Then, 1-step expectation of $\mathbf{r}^\star$ draws a conditional KL divergence:

$$\mathbb{E}_{a\sim\pi(\cdot|s)}\big[\mathbf{r}^\star(s, a) + \mathcal{H}(\pi(\cdot|s))\big] = -\mathrm{D}_{\mathrm{KL}}\big[\pi(\cdot|s)\|\pi_E(\cdot|s)\big] \leq 0 \tag{3}$$

By the property of KL divergence, the above expression is less than or equal to 0 where the equality holds if and only if $\pi(a|s) = \pi_E(a|s), \forall a \in \mathcal{A}$. Since the optimal value function outputs zero, the reward shaping (Ng et al., 1999) of $\mathbf{r}^\star$ is not required. As a result, the optimality of the action is instantaneously determined without consideration of future rewards. It leaves the following remark.

**Remark 4.1.** *For arbitrary discounted rates, $\mathbf{r}^\star(s, a) = \log \pi_E(a|s)$ is the optimally shaped reward function for learning efficiency in the Shannon entropy-regularization.*

From this insight, the log-likelihood of expert policy can be considered as a desirable state-conditioned reward function for the MaxEnt RL. However, oftentimes directly modeling the log density (such as BC methods) is not practical due to the limited number of samples. We can alternatively relaxes likelihood estimation by using a state potential function $\Psi$ (Ng et al., 1999):

$$\mathfrak{R} = \big\{\mathbf{r}\big|\mathbf{r}(s, a) + \gamma\Psi(s') - \Psi(s) = \log \pi_E(a|s), \ \Psi : \mathcal{S} \to \mathbb{R}, \ \forall s, a, s' \in \mathcal{S} \times \mathcal{A} \times \mathcal{S}\big\} \tag{4}$$

Akin to a deterministic case, we formalize a point that Eq. (4) does not change the learning objective.

**Proposition 4.1.** *Let $\mathbf{r}$ be a function satisfying Eq. (4). Then the expected cumulative reward of $\pi$ has following property: $\mathbb{E}_\pi\big[\mathbf{r}(s, a) + \mathcal{H}(\pi(\cdot|s))\big] = -\mathbb{E}_\pi\big[\mathrm{D}_{\mathrm{KL}}\big(\pi(\cdot|s)\|\pi_E(\cdot|s)\big)\big] + \mathbb{E}_{s\sim p_0}\big[\Psi(s)\big]$.*

Note that the term $\mathbb{E}_{s\sim p_0}[\Psi(s)]$ is independent of $\pi$. Compared to the AIRL, the function $\mathbf{r} \in \mathfrak{R}$ can be applied for training arbitrary policies since the overlapping state densities $\log \rho_E(s)/\rho_\pi(s)$ are detached. The subsequent learning of $\mathbf{r}$ provides an projection, which is the closest estimation of $\Pr(a_0, \dots|s_0 = s, \pi_E)$ for the current state.

## 4.3 Handling Finite-Horizon Biases via KL Regularization Assumption

In order to mitigate the terminiation biases depicted in Fig. 1 (a), Kostrikov et al. (2018) highlighted the concept of absorbing states and suggested adding synthetic transitions into the trajectories (see Fig. 1 (b)), which explicitly promotes learning for termination frequencies. In contrast, we focus on another intriguing point that the absorbing state makes all episodes virtually have the properties of infinite horizon. Consider that the expert represents the MaxEnt behavior even when a self-looping state is encountered. Since the action selection can change neither reward nor transition state, the "expert" would represent complete random behavior, which is identical to the uniform distribution.

In the sense that they do not require the explicit manipulation of trajectories, Fig. 1 (c) and (d) demonstrate two possible simpler examples to turn finite-length episodes into infinite-horizon episodes by assuming self-looping terminal states. The two instances are alike for handling the biases as $\mathcal{H}_t = \mathrm{KL}_t + \mathrm{const}$, but the KL regularization has a slight advantage because a terminal

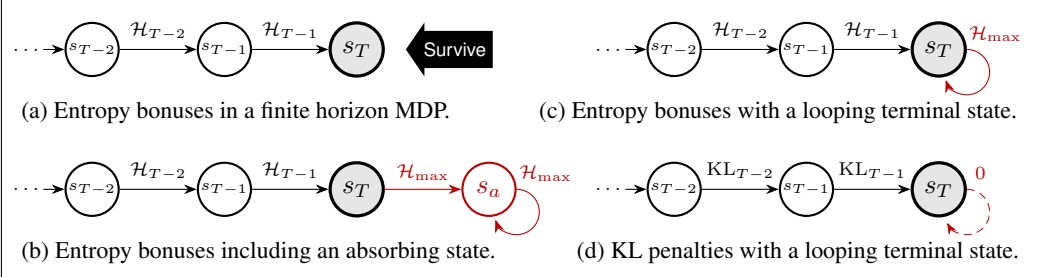

Figure 1: Visualization of entropy regularization methods and biases around terminal states. The red vertices and edges represent absorbing states and maximum entropy action selection. Each regularization is defined as $\mathcal{H}_t = \mathcal{H}\big(\pi(\cdot|s_t)\big)$ and $\mathrm{KL}_t = -\mathrm{D}_{\mathrm{KL}}[\pi(\cdot|s_t)\|p_{\mathrm{unif}}(\cdot)]$, respectively.

state value is always zero instead of $\sum_{k=0}^{\infty}\gamma^k\mathcal{H}_{\mathrm{max}}$. In other words, only switching the regulation scheme from Fig. 1 (a) has eliminated the termination bias. Consequently, the KL penalties can be seamlessly integrated into finite-Horizon MDPs without any pre-processing around terminal states.

With the same reasoning as in Sec. 4.2, the term $\log{\pi_E(a|s)}/u$ can be understood as the optimally shaped reward under the KL regularization. The constant $u$ denotes the likelihood of $p_{\mathrm{unif}}$ (i.e. $u \triangleq p_{\mathrm{unif}}(a), \forall a \in \mathcal{A}$). To this end, we set the learning objective of the shaped reward function:

$$f_{\theta,\psi}(s,a,s') = r_\theta(s,a) + \gamma h_{\bar{\psi}}(s') - h_\psi(s) = \log{\pi_E(a|s)}/u \tag{5}$$

where $r_\theta$ and $h_\psi$ denote reward and potential networks. $h_{\bar{\psi}}$ denotes a target potential network with the identical parameter of $\psi$, yet the gradient computation is disconnected while training, which is analogous to constructing target value networks in the deep RL domain (Mnih et al., 2015). For every terminal state, we fixed the output to the optimal solution: $r_\theta(s_T,a) = h_\psi(s_T) = 0, \forall a \in \mathcal{A}$.

## 5 CAUSAL ADVERSARIAL INVERSE REINFORCEMENT LEARNING

### 5.1 A DUAL DISCRIMINATOR ARCHITECTURE

The methodology is grounded on the findings that an AIL discriminator also contains an EBM, and the property that an EBM on joint densities can be decomposed into multiple EBMs (Zhao et al., 2016; Che et al., 2020; Azadi et al., 2018). Suppose that the GAIL discriminator is nearly optimal:

$$D(s,a) = \frac{1}{1 + \exp\big(-d(s,a)\big)} \approx \frac{\rho_E(s,a)}{\rho_E(s,a) + \rho_\pi(s,a)}$$

where $d(s,a)$ denotes the logit of $D(s,a)$. We disentangle the logit function to the following form:

$$d(s,a) \approx \log\frac{\rho_E(s,a)}{\rho_\pi(s,a)} = \log\frac{\rho_E(s)}{\rho_\pi(s)} + \log\frac{\pi_E(a|s)}{\pi(a|s)} \tag{6}$$

and then we have two log-ratios of state occupancy measures and policy distributions.

The model substitutes the log-ratio of state occupancy measures using a state-only discriminator $D_\varphi$ with a nearly optimal logit score of $d_\varphi(s) \approx \log(\rho_E(s)/\rho_\pi(s))$. The role of $D_\varphi$ is to nullify the difference between state densities by pre-applying it. We propose an architecture of discriminator:

$$D_{\theta,\psi}(s,a,s') = \frac{\exp\big[f_{\theta,\psi}(s,a,s')\big]}{\exp\big[f_{\theta,\psi}(s,a,s')\big] + \exp\big[-d_\varphi(s) + \log\frac{\pi_\phi(a|s)}{u}\big]} \tag{7}$$

where $d_\varphi(s)$ and $\pi_\phi(a|s)$ are pre-computed. Using $D_\varphi$ and $\pi_\phi$ as scaffolds, the shaped reward function $f_{\theta,\psi}$ converges to Eq. (5) and becomes specialized in evaluating conditional decisions, where learning with the reward function correctly projects to the optimal policy.

The discriminators $D_\varphi$ and $D_{\theta,\psi}$ are trained for maximizing the following objectives, respectively:

$$\mathcal{J}(D_\varphi) = \mathbb{E}_{\pi_E}\big[\log D_\varphi(s)\big] + \mathbb{E}_{\pi_\phi}\big[\log(1 - D_\varphi(s))\big],$$
$$\mathcal{J}(D_{\theta,\psi}) = \mathbb{E}_{\pi_E}\big[\log D_{\theta,\psi}(s,a,s')\big] + \mathbb{E}_{\pi_\phi}\big[\log(1 - D_{\theta,\psi}(s,a,s'))\big] - \lambda\mathbb{E}_{\pi_\phi,\pi_E}\big[\|\chi_\psi(s,s')\|_2^2\big],$$

where $\chi_\psi(s, s') = \gamma h_{\bar\psi}(s') - h_\psi(s)$ denotes the shaping function. To make the $r_\theta$ to be close to the case $\log \pi_E(a|s)/u$, we regularize the function by minimizing squared L2-norm with $\lambda \in \mathbb{R}^+$. The regularization on the shaping function eventually makes it converge to zero, but it achieves relatively stable results than regularizing $\|h_\psi(s)\|_2^2$. The algorithm provides an IRL reward as $\alpha \cdot r_\theta(s, a)$; ideally, the performance is invariant to $\alpha$, if all the processes share the same temperature. In some benchmarks, the reward function is constrained by using the softmax activation function. As the terminal states get the highest entropy bonuses, this constraint does not exploit awareness of terminal states. In some benchmarks it has the practical advantage of preventing overly pessimistic rewards when IRL is not sufficiently trained. We defer additional implementation details to Appendix D.

## 5.2 ANALYSES ON THE CAUSAL AIRL ALGORITHM

**Entropy-regularized IRL**. We draw a connection between entropy-regularized policy gradient algorithms and our method as an adversarial training method for the policy network.

**Proposition 5.1.** *If Eq. (5) is satisfied, the following equality holds.*

$$\nabla_\phi \mathbb{E}_{\pi_\phi}\big[\mathrm{D}_{\mathrm{KL}}\big(\pi_\phi(\cdot|s)\big\|\pi_E(\cdot|s)\big)\big] = -\mathbb{E}_{\pi_\phi}\big[Q^{\pi_\phi}(s,a)\nabla_\phi \log \pi_\phi(a|s) + \nabla_\phi \mathcal{H}(\cdot|s)\big], \qquad (8)$$

*where* $Q^\pi(s,a) \triangleq r_\theta(s,a) + \mathbb{E}\big[\sum_{t=1}^\infty \gamma^t(r_\theta(s_t, a_t) - \log \pi_\phi(a_t|s_t))\big|s_0 = s, a_0 = a, \pi\big]$.

The proposition again shows the strong relationship between entropy-regularized RL and AIL. Unlike deterministic RL, the policy of entropy-regularized RL is proved to be converged to a unique fixed point in a regularized condition (Geist et al., 2019; Yang et al., 2019). Thus, we deduce that the learning scheme of CAIRL leads the policy to be converged to the fixed point of the expert.

**Application to Transfer Learning**. Suppose that the expert resides in another MDP: $M_E = (\mathcal{S}, \mathcal{A}, \mathcal{P}_E, r, \bar p_0, \gamma)$. Extending our formulation, we can induce that the function $f_{\theta,\psi}(s, a, s')$ would converge to $\log\big(\frac{\mathcal{P}_E(s'|s,a)}{\mathcal{P}(s'|s,a)} \cdot \frac{\pi_E(a|s)}{u}\big)$, and the expectation of the function with the KL penalty gives

$$\mathbb{E}_{a\sim\pi(\cdot|s), s'\sim\mathcal{P}(\cdot|s,a)}\big[f_{\theta,\psi}(s, a, s') - \mathrm{D}_{\mathrm{KL}}(\pi(\cdot|s)\|p_{\mathrm{unif}}(\cdot))\big] = -\mathrm{D}_{\mathrm{KL}}\big[p_{\pi_\phi}(a, s'|s)\big\|p_E(a, s'|s)\big], \tag{9}$$

where $p_{\pi_\phi}(a, s'|s)$ and $p_E(a, s'|s)$ denote the agent's and the expert's conditional probability distributions of action and transition state, and learning with $r_\theta(s, a)$ also promotes the identical effect. If the distributions $\bar p_0$ and $\mathcal{P}_E$ are different, the optimal behavior has to adapt to the gap between domains. The energy-based reward function $r_\theta(s, a)$ is robust to dynamics misalignment as it learns the conditional joint distribution of actions and transition states, namely $\mathrm{D}_{\mathrm{KL}}\big[\Pr(a_0, s_1, \ldots|s_0 = s, \pi)\big\|\Pr(a_0, s_1, \ldots|s_0 = s, \pi_E)\big]$.

**Objective of Potential Networks**. The reward shaping can be viewed as a value iteration scheme. We relate Eq. (5) to an entropy-regularized operator called *mellowmax* defined as $mm_\alpha(X) = \log\big(\frac{1}{n}\sum_{i=1}^n \exp(\alpha x_i)\big)/\alpha$, and show that $r_\theta$ and $h_\psi$ satisfy the mellowmax Bellman optimality.

**Proposition 5.2.** *If* $f_{\theta,\psi}(s, a, s') = \log \pi_E(a|s)/u$ *for all states, actions, and transition states, then*

$$h_\psi(s) = \log\bigg[u \cdot \int_{\mathcal{A}} \exp\big\{r_\theta(s,a) + \gamma\mathbb{E}_{s'|s,a}\big[h_\psi(s')\big]\big\}\,\mathrm{d}a\bigg]$$

*Thus* $h_\psi$ *corresponds to the mellowmax optimal value function of* $r_\theta$ *with* $\alpha = 1$.

According to analyses of Asadi & Littman (2017), the standard softmax operator may lead the value learning to multiple fixed-points when $\gamma \approx 1$. As the mellowmax is a non-expansion operator, the use of the KL penalties achieves relatively stable potential function optimization.

## 6 EXPERIMENTAL RESULTS

Our experiments aim to understand CAIRL and verify the effectiveness of the algorithm in the sense of our claims. We evaluate our approach on three topics, whose settings are motivated by the previous works (Haarnoja et al., 2017; 2018; Fu et al., 2017). For the RL algorithms, we implemented an algorithm based on `OpenAI-PPO2` (Schulman et al., 2017) and use the KL regularization in order to eliminate the survival biases in MaxEnt RL as addressed in Sec. 4.3.

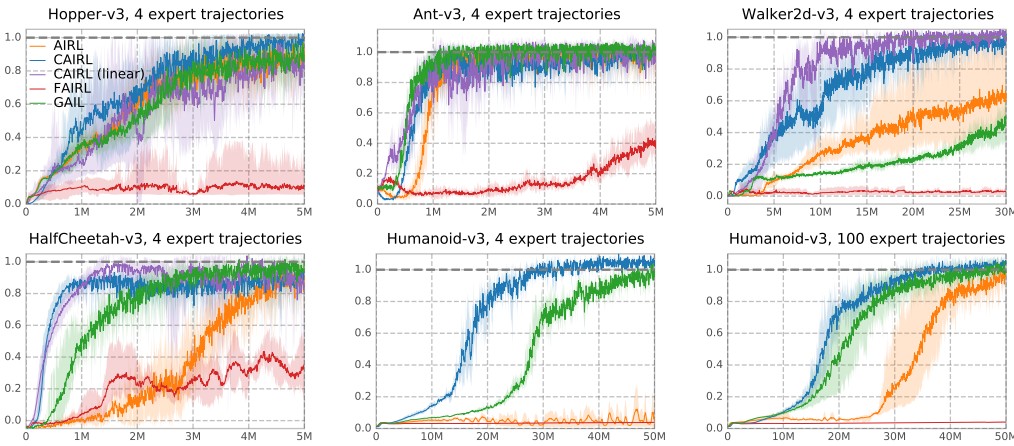

Figure 2: IRL trajectories of 2D Multi-goal environments.   Figure 3: Ant robot trajectories.

Figure 4: Training curves of stochastic policies on imitation learning benchmarks.

**Multi-Modal Behavior**. The first experiment setting is a multi-goal environment. From the initial agent position, four goals are located at the four cardinal directions. While a deterministic policy commits to a single goal at the earliest attempt, we hypothesize that the optimal MaxEnt policy distribution represents a multi-modal behavior, which is reaching all the four goals at the same rate. We evaluated algorithms on two settings. In the 2D setting, the agent is a point mass. The ground-truth reward function is defined as the difference between Gaussian mixture model values of points: $\texttt{GMM}(\mathbf{x}_{t+1}) - \texttt{GMM}(\mathbf{x}_t)$ where $\mathbf{x}_t$ is a 2D representation of state. In the 3D setting, the agent is a simulated robot where its state is defined by the position and its joint values. The detailed setting and expert of each of the environments are provided in Appendix C.1.

Fig. 2 visualizes trajectories obtained by the IRL algorithms. The symmetric multi-goal environment experiment (left) demonstrates that CAIRL is capable of recovering a multi-modal policy resulting to reach all goals. For evaluating survival bias handling, we implemented a similar task, called an asymmetric multi-goal environment, where the right side goal is located substantially further than others (right). Table 2 displays an ablation study that shows episode time steps and ratios that an IRL agent reach the goal in the right-side. The result shows that CAIRL can induce more uniform multi-modal distribution than AIRL variants and that the algorithm is robust to the survival bias. Especially, the CAIRL algorithm the both activation functions achieved

Table 2: Statistics of CAIRL and AIRL in the asymmetric multi-goal environment.

|  | Ep. Timestep | Right-side |
|---|---|---|
| Expert | 42.61 $\pm_{18.51}$ | 0.25 |
| AIRL | 70.11 $\pm_{80.35}$ | 0.24 |
| AIRL+absorb | 87.22 $\pm_{108.0}$ | 0.33 |
| AIRL+softplus | 52.33 $\pm_{52.08}$ | 0.31 |
| CAIRL+no KL | 40.9 $\pm_{19.3}$ | 0.23 |
| CAIRL+linear | 41.07 $\pm_{17.47}$ | 0.24 |
| CAIRL+softplus | 42.05 $\pm_{19.46}$ | 0.25 |

high performance; the difference between two cases was not significant. Fig. 3 shows a result that CAIRL outperforms AIRL in the quality of generated trajectories. CAIRL searched all the four goals with the robot, which prominently shows that our algorithm can reconstruct rewards from multi-modal policies in the complex control tasks.

**Imitation Learning**. The second experiment is the imitation learning tasks of continuous domains from the Gym benchmark suite (Brockman et al., 2016). We evaluated our algorithm on five challenging tasks, including Humanoid benchmark with 21 action dimensions.

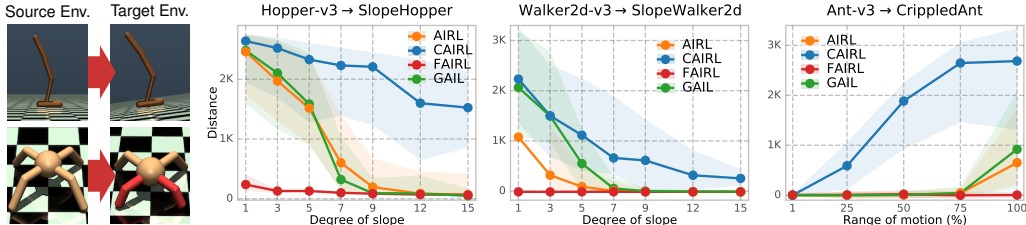

Figure 5: Illustrations of transfer learning and total distance traveled by transfer learning agents.

In Fig. 4, training curves are visualized with the results of five individual runs where the scores are rescaled with respect to the expert score, and the shaded regions represent the minimum and maximum scores. Since expert policies are uni-modal Gaussian policies, we restricted the number of trajectories to 4 except the Humanoid task. In the figure, CAIRL algorithms (blue lines) shows the fast learning speed in the early phase of training and at least the second-best results for all provided experiments. CAIRL clearly outperforms previous methods in the Hopper-v3, Walker2d-v3, and the Humanoid-v3 tasks. These tasks are considered to be challenging due to the harsh termination condition. We highlight that CAIRL with linear activation (purple lines) also shows competitive performance in the conducted experiments. In our experiments, FAIRL showed the lowest performance. We suspect that the FAIRL reward formulation $r(s,a) = \exp(d(s,a)) \cdot (-d(s,a))$ sometimes give too much penalty for a single step reward, even when the proposed clipping method is applied.

**Transfer Learning**. The third experiment is the transfer learning tasks. The setup is inspired by Fu et al. (2017), but the settings are quite different. We trained each IRL network in the target (test) environment. To simply put, our experiment aims to measure the flexibility of the reward learning process. The formulation was designed to show that knowledge transfer requires adaptation. We wanted to find out whether each algorithm provides helpful rewards without pretraining. Also, it is natural to think that only a chunk of trajectories is available in a realistic problem. We implemented three transfer learning tasks called SlopeHopper, SlopeWalker2d, and CrippledAnt. In the Slope-Hopper and SlopeWalker tasks, the agent has the same configuration with the original 3D models but the ground is tilted by certain degree ranges to $[1, 15]$. In the CrippledAnt task, the robot has noticeably shorter forelegs as colored red in Fig. 5. We additionally restricted the joint angles of forelegs in the range of $\{.01, .25, .50, .75, 1.0\}$ compared to the original model.

Given the expert trajectories from the source environments, the results of transfer learning tasks are shown in Fig. 5. For each task, we repeated all runs 5 times and report the results which are averaged over scores from the last 0.5 million training steps. In transfer learning setting, CAIRL outperformed other algorithms considerably that it achieved the highest performance for every experiment. The results imply that the algorithm extracted informative rewards for new environments with different dynamics, such that our reward acquisition methods are robust with the variation between tasks. Other algorithms failed considerably; it empirically validate one of our hypothesis that the cumulative sum of state occupancy measures hiders robust learning when domain shift happens. The experiments have verified that with the proper consideration of temporal dependencies, the AIL algorithms could be extended towards transfer learning problems.

## 7 CONCLUSION

In this paper, we have proposed a causal AIRL algorithm that recovers a robust reward function. We have provided theoretical analyses, including reward shaping in entropy-regularized MDPs, and the connection between adversarial learning and energy-based RL. We have proposed a novel dual discriminator architecture, which learns a reward and a value function of regularized Bellman optimality equations. Our model can efficiently disentangle biases originated from state occupancy and terminal states. We have verified that the proposed IRL method has clear advantages over AIRL for learning multi-modal behaviors and handling termination biases. The proposed method recovers state-conditioned rewards, which has advantages over AIL algorithms in terms of the robustness of imitation performance in challenging continuous domains. Furthermore, the proposed method outperformed other methods in domain adaptation in the transfer learning experiments.

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

## A   PROOFS

**Proposition 4.1.** *Let* **r** *be a function satisfying Eq. (4). Then the expected cumulative reward of $\pi$ has following property:* $\mathbb{E}_\pi\big[\mathbf{r}(s,a) + \mathcal{H}(\pi(\cdot|s))\big] = -\mathbb{E}_\pi\big[\mathrm{D}_{\mathrm{KL}}\big(\pi(\cdot|s)\big\|\pi_E(\cdot|s)\big)\big] + \mathbb{E}_{s\sim p_0}\big[\Psi(s)\big].$

*Proof.* $\rho_\pi(s)$ denotes the state-only occuany measure of state $s$.

$$
\begin{aligned}
\mathbb{E}_\pi[\mathbf{r}(s,a) + \mathcal{H}(\pi(\cdot|s))] &= \mathbb{E}\left[\sum_{t=0}^\infty \gamma^t(\mathbf{r}(s_t,a_t) - \log\pi(a_t|s_t))\right] \\
&= \mathbb{E}\left[\sum_{t=0}^\infty \gamma^t(\log\pi_E(a_t|s_t) - \log\pi(a_t|s_t) - \gamma\Psi(s_{t+1}) + \Psi(s_t))\right] \\
&= \mathbb{E}\left[\sum_{t=0}^\infty \gamma^t(\log\frac{\pi_E(a_t|s_t)}{\pi(a_t|s_t)}) - \sum_{t=0}^\infty \gamma^{t+1}\Psi(s_{t+1}) + \sum_{t=0}^\infty \gamma^t\Psi(s_t)\right] \\
&= \mathbb{E}\left[\sum_{t=0}^\infty \gamma^t(\log\frac{\pi_E(a_t|s_t)}{\pi(a_t|s_t)}) - \sum_{t=1}^\infty \gamma^t\Psi(s_t) + \sum_{t=0}^\infty \gamma^t\Psi(s_t)\right] \\
&= \mathbb{E}\left[\sum_{t=0}^\infty \gamma^t(\log\frac{\pi_E(a_t|s_t)}{\pi(a_t|s_t)}) + \Psi(s_0)\right] \\
&= \int_{\mathcal{S}}\int_{\mathcal{A}} \rho_\pi(s,a)\left[\log\frac{\pi_E(a|s)}{\pi(a|s)}\right]\mathrm{d}a\,\mathrm{d}s + \int_{\mathcal{S}} p_0(s)\Psi(s)\,\mathrm{d}s \\
&= \int_{\mathcal{S}} \rho_\pi(s)\int_{\mathcal{A}} \pi(a|s)\left[\log\frac{\pi_E(a|s)}{\pi(a|s)}\right]\mathrm{d}a\,\mathrm{d}s + \mathbb{E}_{s\sim p_0}[\Psi(s)] \\
&= -\int_{\mathcal{S}} \rho_\pi(s)(\mathrm{D}_{\mathrm{KL}}(\pi(\cdot|s)\|\pi_E(\cdot|s)))\,\mathrm{d}s + \mathbb{E}_{s\sim p_0}[\Psi(s)] \\
&= -\int_{\mathcal{S}} \rho_\pi(s)\int_{\mathcal{A}} \pi(a|s)(\mathrm{D}_{\mathrm{KL}}(\pi(\cdot|s)\|\pi_E(\cdot|s)))\,\mathrm{d}a\,\mathrm{d}s + \mathbb{E}_{s\sim p_0}[\Psi(s)] \\
&= -\mathbb{E}_\pi\big[\mathrm{D}_{\mathrm{KL}}\big(\pi(\cdot|s)\big\|\pi_E(\cdot|s)\big)\big] + \mathbb{E}_{s\sim p_0}\big[\Psi(s)\big]
\end{aligned}
$$

Therefore, the reward function $\mathbf{r}(s,a)$ also provides the same global objective as the likelihood $\log\pi_E(a|s)$.   □

**Proposition 5.1.** *If all critic functions are optimal, the following equality holds.*

$$
\nabla_\phi\mathbb{E}_{\pi_\phi}\big[\mathrm{D}_{\mathrm{KL}}\big(\pi_\phi(\cdot|s)\big\|\pi_E(\cdot|s)\big)\big] = -\mathbb{E}_{\pi_\phi}\big[Q^{\pi_\phi}(s,a)\nabla_\phi\log\pi_\phi(a|s) + \nabla_\phi\mathcal{H}(\cdot|s)\big],
$$

*where* $Q^\pi(s,a) \triangleq r_\theta(s,a) + \mathbb{E}\big[\sum_{t=1}^\infty \gamma^t(r_\theta(s_t,a_t) - \log\pi_\phi(a_t|s_t))\big|s_0 = s, a_0 = a, \pi\big].$

*Proof.* By the policy gradient theorem Sutton & Barto (2018), we can derive the gradient of $\phi$ when initial state is $s$.

$$
\begin{aligned}
-\nabla_\phi\mathbb{E}_{\pi_\phi}\Big[\mathrm{D}_{\mathrm{KL}}\big[\pi_\phi(\cdot|s)\big\|\pi_E(\cdot|s)\big]\Big|s_0 = s\Big] &= \nabla_\phi(\mathbb{E}_{\pi_\phi}\big[r_\theta(s,a) - \log\pi_\phi(a|s)\big|s_0 = s\big] + const.) \\
&= \nabla_\phi\mathbb{E}\left[\sum_{t=0}^\infty \gamma^t\big(r_\theta(s_t,a_t) - \log\pi_\phi(a_t|s_t)\big)\Big|s_0 = s\right] \\
&= \nabla_\phi V^{\pi_\phi}(s)
\end{aligned}
$$

where $V^\pi(s) \triangleq \mathbb{E}\big[\sum_{t=0}^\infty \gamma^t\big(r_\theta(s_t,a_t) - \log\pi_\phi(a_t|s_t)\big)\big|s_0 = s, \pi\big].$ By the product rule, we get

$$
\begin{aligned}
\nabla_\phi V^{\pi_\phi}(s) &= \int_{\mathcal{A}} \nabla_\phi\pi_\phi(a|s)\Big(Q^\pi(s,a) - \log\pi_\phi(a|s)\Big) + \pi_\phi(a|s)\nabla_\phi\Big(Q^\pi(s,a) - \log\pi_\phi(a|s)\Big)\,\mathrm{d}a \\
&= \int_{\mathcal{A}} \nabla_\phi\pi_\phi(a|s)Q^\pi(s,a) + \pi_\phi(a|s)\nabla_\phi\mathcal{H}(\cdot|s) + \pi_\phi(a|s)\int_{\mathcal{S}} \gamma\mathcal{P}(s'|s,a)\nabla_\phi V^{\pi_\phi}(s')\,\mathrm{d}s'\,\mathrm{d}a
\end{aligned}
$$

By repeatedly unrolling $\nabla_\phi V^\pi(s_t)$, we can derive the following form:

$$\nabla_\phi V^\pi(s) = \int_{\mathcal{S}} \sum_{k=0}^{\infty} \gamma^k \Pr(s \to x, k, \pi_\phi) \int_{\mathcal{A}} \left( \nabla_\phi \pi_\phi(a|x) Q^\pi(x, a) + \pi_\phi(a|x) \nabla_\phi \mathcal{H}(\cdot|x) \right) \mathrm{d}a \, \mathrm{d}x$$

where $\Pr(s \to x, k, \pi)$ is the probability of transitioning from state $s$ to state $x$ in $k$ steps under policy $\pi$. Thus, using the log derivative trick we can derive the rest of equations as follows

$$\nabla_\phi \mathbb{E}_{\pi_\phi} \left[ D_{\mathrm{KL}} \left[ \pi_\phi(\cdot|s) \| \pi_E(\cdot|s) \right] \right] = \int_{\mathcal{S}} p_0(s) \nabla_\phi \mathbb{E}_{\pi_\phi} \left[ D_{\mathrm{KL}} \left[ \pi_\phi(\cdot|s) \| \pi_E(\cdot|s) \right] \Big| s_0 = s \right] \mathrm{d}s$$

$$= -\int_{\mathcal{S}} \sum_{t=0}^{\infty} \gamma^t Pr(s_t = s|\pi) \int_{\mathcal{A}} \left( \nabla_\phi \pi_\phi(a|s) Q^\pi(s, a) + \pi_\phi(a|s) \nabla_\phi \mathcal{H}(\cdot|s) \right) \mathrm{d}a \, \mathrm{d}s$$

$$= -\int_{\mathcal{S}} \rho_{\pi_\phi}(s) \int_{\mathcal{A}} \left( \nabla_\phi \pi_\phi(a|s) Q^\pi(s, a) + \pi_\phi(a|s) \nabla_\phi \mathcal{H}(\cdot|s) \right) \mathrm{d}a \, \mathrm{d}s$$

$$= -\int_{\mathcal{S}} \rho_{\pi_\phi}(s) \int_{\mathcal{A}} \pi_\phi(a|s) \Big( (Q^\pi(s, a) \nabla_\phi \log \pi_\phi(a|s) + \nabla_\phi \mathcal{H}(\cdot|s) \Big) \mathrm{d}a \, \mathrm{d}s$$

$$= -\mathbb{E}_{\pi_\phi} \left[ Q^\pi(s, a) \nabla_\phi \log \pi_\phi(a|s) + \nabla_\phi \mathcal{H}(\cdot|s) \right]$$

$\square$

**Proposition 5.2.** *If $f_{\theta,\psi}(s, a, s') = \log \pi_E(a|s)/u$ for all states, actions, and transition states, then*

$$h_\psi(s) = \log \left[ u \cdot \int_{\mathcal{A}} \exp\{ r_\theta(s, a) + \gamma \mathbb{E}_{s'|s,a} \left[ h_\psi(s') \right] \} \, \mathrm{d}a \right]$$

*Thus $h_\psi$ corresponds to the optimal value function of the mellowmax regularizer with $\alpha = 1$.*

*Proof.*

$$r_\theta(s, a) + \gamma h_\psi(s') - h_\psi(s) = \log \pi_E(a|s) - \log u \quad \forall s' \in \mathcal{S}$$

$$\implies r_\theta(s, a) + \gamma \mathbb{E}_{s'|s,a} [h_\psi(s')] - h_\psi(s) = \log \pi_E(a|s) - \log u$$

$$\implies u \exp(r_\theta(s, a) + \gamma \mathbb{E}_{s'|s,a} [h_\psi(s')] - h_\psi(s)) = \pi_E(a|s)$$

$$\implies u \frac{\exp(r_\theta(s, a) + \gamma \mathbb{E}_{s'|s,a} [h_\psi(s')])}{\exp(h_\psi(s))} = \pi_E(a|s) \tag{10}$$

$$\implies \int_{\mathcal{A}} u \cdot \frac{\exp(r_\theta(s, a) + \gamma \mathbb{E}_{s'|s,a} [h_\psi(s')])}{\exp(h_\psi(s))} \, \mathrm{d}a = \int_{\mathcal{A}} \pi_E(a|s) \, \mathrm{d}a$$

$$\implies h_\psi(s) = \log \left[ u \cdot \int_{\mathcal{A}} \exp\{ r_\theta(s, a) + \gamma \mathbb{E}_{s' \sim \mathcal{P}(\cdot|s,a)} \left[ h_\psi(s') \right] \} \, \mathrm{d}a \right]$$

$\square$

# B    DISCUSSIONS WITH MAXENT IRL METHODS

In this section, we address similarities between frameworks of the maximum causal entropy IRL and the causal AIRL.

## B.1    MAXIMUM CAUSAL ENTROPY FRAMEWORK

Following the notation of Kramer (1998), the causal entropy can be defined as

$$\mathcal{H}(\mathbf{A}^T \| \mathbf{S}^T) \triangleq \mathbb{E}_{A,S}[-\log P(\mathbf{A}^T \| \mathbf{S}^T)] = \sum_{t=1}^{T} \mathcal{H}(A_t | \mathbf{S}_{1:t}, \mathbf{A}_{1:t-1})$$

The objective of maximum causal entropy IRL can be formulated by the following optimization Ziebart et al. (2010):

$$\underset{P(A_t|S_t)}{\arg\max} \mathcal{H}(\mathbf{A}^T||\mathbf{S}^T)$$

$$\text{such that: } \mathbb{E}_{p(S,A)}[\mathcal{F}(S,A)] = \mathbb{E}_{\hat{p}(S,A)}[\mathcal{F}(S,A)]$$

$$\forall s_t \quad \sum_{a_t} P(A_t|\mathbf{S}_{1:t}, \mathbf{A}_{1:t-1}) = 1$$

where the overall dynamics $P(\mathbf{S}^T||\mathbf{A}^{T-1}) \triangleq \prod_{t=1}^{T} P(S_t|\mathbf{S}_{t-1}, \mathbf{A}_{t-1})$ are given.

According to Ziebart et al. (2010), the distribution satisfying the constrained optimization problem can be defined as a recurrence relation:

$$p_\theta(a_t|s_t) = \frac{\beta(s_t, a_t)}{\beta(s_t)}$$

$$\log \beta(s_t, a_t) = \theta^\mathsf{T} \mathcal{F}_{s_t, a_t} + \sum_{s_{t+1}} p(s_{t+1}|s_t, a_t) \log \beta(s_{t+1})$$

$$\log \beta(s_t) = \log \sum_{a_t} \beta(s_t, a_t) = \underset{a_t}{\text{softmax}} \log \beta(s_t, a_t)$$

With the perspective of energy-based RL, we draw the following implications from the solution:

- $p_\theta(a_t|s_t)$ can be understood as the unique fixed point of MaxEnt framework, which can be achieved by the optimization via soft RL algorithms.
- $r_\theta(s_t, a_t) = \theta^\mathsf{T} \mathcal{F}_{s_t, a_t}$ can be understood as a reward function which is subject to the linear constraints on feature function.
- $\log \beta(s_t, a_t)$ and $\log \beta(s_t)$ can be understood as the (optimal) soft values satisfying the Bellman equation.

## B.2 VANILLA AIRL REWARD FUNCTION DOES NOT FORMULATE AN EBM

Adversarial Inverse Reinforcement Learning (AIRL) (Fu et al., 2017) is a well-known IRL method that applies an adversarial architecture to solve the IRL problem. Formally, AIRL constructs the discriminator as

$$D(s, a) = \frac{\exp\{f(s, a)\}}{\exp\{f(s, a)\} + \pi(a|s)} \tag{11}$$

This is highly motivated by the former GAN-GCL work (Finn et al., 2016b), which proposes that one can apply GAN to train the discriminator as

$$D(\tau) = \frac{\frac{1}{Z} \exp(c(\tau))}{\frac{1}{Z} \exp(c(\tau)) + \pi(\tau)}. \tag{12}$$

where $c(\tau)$ denotes the cost function for trajectories. From the AIL formulation, AIRL uses a surrogate reward as

$$r(s, a) = \log D(s, a) - \log(1 - D(s, a)) = f(s, a) - \log \pi(a|s), \tag{13}$$

where $f(s, a) = \log(\rho_E(s, a)/\rho_\pi(s))$ and the overall reward can be seen as an entropy-regularized reward function. If $\gamma = 0$, it is evident that AIRL is identical to the standard adversarial learning without temporal sequences. As a result, it makes sense to directly optimize the policy by taking the energy model as the target policy instead of the reward function, which leads to the optimal solution as:

$$\pi^\star(a|s) = \frac{\exp(Q^\pi(s, a))}{\sum_{a'} \exp(Q^\pi(s, a'))} = \frac{\exp(f(s, a))}{\sum_{a'} \exp(f(s, a'))} = \frac{\rho_{\pi_E}(s, a)}{\sum_{a'} \rho_{\pi_E}(s, a')}, \quad \gamma = 0.$$

By the fundamental property of occupancy measure (Theorem 2 of Syed et al. 2008). However, for the general case, the projection of Q-value function cannot be close to $\pi_E$ like the standard probabilitic inference using energy-based models.

### B.3 Causal Adversarial Inverse Reinforcement Learning Formulation

As a direct interpretation of the maximum causal entropy framework, the goal of CAIRL can be seen as training a parameterized distribution over the expert trajectories as the following maximum-likelihood objective:

$$\max_\theta \mathcal{J}(\theta) = \max_\theta \mathbb{E}_{\pi_E}[\log p_\theta(a|s)], \tag{12}$$

where the conditional distribution $p_\theta(a_t|s_t)$ is parameterized as $p_\theta(a_t|s_t) \propto \exp\left(Q_\theta(s_t, a_t)\right)$. We can compute the gradient with respect to $\theta$ as follows:

$$\begin{aligned}
\frac{\partial}{\partial\theta}\mathcal{J}(\theta) &= \mathbb{E}_{\pi_E}\left[\frac{\partial}{\partial\theta}\log p_\theta(a|s)\right] \\
&= \mathbb{E}_{\pi_E}\left[\frac{\partial}{\partial\theta}Q_\theta(s,a) - \frac{\partial}{\partial\theta}\log Z_\theta(s)\right] \\
&= \mathbb{E}_{\pi_E}\left[\frac{\partial}{\partial\theta}Q_\theta(s,a)\right] - \mathbb{E}_{\pi_E}\left[\mathbb{E}_{a'\sim p_\theta(\cdot|s)}\left[\frac{\partial}{\partial\theta}Q_\theta(s,a')\right]\right],
\end{aligned} \tag{13}$$

where $Z_\theta(s)$ is the partition function normalizes the distribution $p_\theta$. Since sampling with $\pi_E$ and $p_\theta$ is difficult, we can think of substituting the formulation with the following adversarial learning form:

$$\mathbb{E}_{\pi_E}\left[\frac{\partial}{\partial\theta}r_\theta(s,a)\right] - \mathbb{E}_\pi\left[\frac{\partial}{\partial\theta}r_\theta(s,a)\right]. \tag{14}$$

Nevertheless the formulation in Eq. (14) is similar with a standard adversarial framework with i.i.d. data, the approximation is not a safe choice because of unbounded divergences among $\pi$, $p_\theta$, and $\pi_E$. Therefore, appropriate adversarial learning in sequential decision problems is essentially different to GANs for joint distribution matching.

In CAIRL, we replace the reward learning objective with training a logistic discriminator:

$$D_\theta(s,a) = \frac{\exp[f_\theta(s,a)]}{\exp[f_\theta(s,a)] + \kappa(s,a)},$$

where $\kappa(s,a) = \frac{\rho_\pi(s,a)}{\rho_E(s)\cdot u}$. The objective of the discriminator is to maximize the generalized Jensen-Shannon divergence between of the generated samples:

$$\mathcal{J}(D_\theta) = \mathbb{E}_{\pi_E}[\log D_\theta(s,a)] + \mathbb{E}_\pi[\log(1 - D_\theta(s,a))]$$

For the discriminators, the objective can be written as follows:

$$\begin{aligned}
\mathcal{J}(D_\theta) &= \mathbb{E}_{\pi_E}[\log D_\theta(s,a)] + \mathbb{E}_\pi[\log(1 - D_\theta(s,a))] \\
&= \mathbb{E}_{\pi_E}\left[\log\frac{\exp[f_\theta(s,a)]}{\exp[f_\theta(s,a)] + \kappa(s,a)}\right] + \mathbb{E}_\pi\left[\log\frac{\kappa(s,a)}{\exp[f_\theta(s,a)] + \kappa(s,a)}\right] \\
&= \mathbb{E}_{\pi_E}\left[f_\theta(s,a)\right] + \mathbb{E}_\pi\left[\kappa(s,a)\right] - 2\cdot\mathbb{E}_{\tilde\pi}\left[\log\left\{\exp[f_\theta(s,a)] + \kappa(s,a)\right\}\right],
\end{aligned}$$

where the operator $\mathbb{E}_{\tilde\pi}$ denote the expectation using an occupancy measure $\rho_{\tilde\pi}(s,a) = \frac{\rho_E(s,a)+\rho_\pi(s,a)}{2}$. Taking the derivative with respect to $\theta$,

$$\begin{aligned}
\frac{\partial}{\partial\theta}\mathcal{J}(D_\theta) &= \mathbb{E}_{\pi_E}\left[\frac{\partial}{\partial\theta}f_\theta(s,a)\right] - 2\cdot\mathbb{E}_{\tilde\pi}\left[\frac{\exp[f_\theta(s,a)]}{\exp[f_\theta(s,a)] + \kappa(s,a)}\frac{\partial}{\partial\theta}f_\theta(s,a)\right] \\
&= \mathbb{E}_{\pi_E}\left[\frac{\partial}{\partial\theta}f_\theta(s,a)\right] - \mathbb{E}_{\tilde\pi}\left[\frac{\rho_E(s)\hat p_\theta(a|s)}{\frac12\rho_E(s)\hat p_\theta(a|s) + \frac12\rho_\pi(s,a)}\frac{\partial}{\partial\theta}f_\theta(s,a)\right] \\
&= \mathbb{E}_{\pi_E}\left[\frac{\partial}{\partial\theta}f_\theta(s,a)\right] - \mathbb{E}_{\pi_E}\left[\sum_{a'\in\mathcal{A}}\hat p_\theta(a'|s)\cdot\delta(s,a')\cdot\frac{\partial}{\partial\theta}f_\theta(s,a')\right],
\end{aligned}$$

where

$$\hat p_\theta(a|s) = u\cdot\exp(f_\theta(s,a)) \quad\text{and}\quad \delta(s,a) = \frac{\rho_E(s,a) + \rho_\pi(s,a)}{\rho_E(s)\hat p_\theta(a|s) + \rho_\pi(s,a)}.$$

The optimal $\theta$ can be easily found by considering $\frac{\partial}{\partial\theta}\mathcal{J}(D_\theta) = 0$, in condition of $\hat p_\theta(a|s) = \pi_E(a|s)$ with $\delta(s,a) = 1$ for all states and actions. Moreover, for the right hand side of the expression, we can draw the following properties of $\delta$:

- If $\rho_\pi(s, a) \gg \rho_E(s, a)$, meaning the supports are disjoint, $\delta(s, a) \approx 1$.

- If $\rho_\pi(s, a) \approx \rho_E(s, a)$, where $\hat{p}_\theta(a|s)$ is normally closer to $\pi_E$ than $\pi$, $\delta(s, a) \approx 1$.

Therefore, the expression matches the Eq. (13). CAIRL approximates the maximum causal entropy framework without exhaustively computing the recursive equation, and also recovers a plausible reward function on both supports of $\rho_\pi$ and $\rho_E$.

## C  EXPERIMENTAL DETAILS

### C.1  MUTL-GOAL ENVIRONMENTS

- 2D Point Environment: Let the 2D coordinate denote the position of a point mass on the environment. The agent is generated according to the normal distribution $\mathcal{N}(\mathbf{0}, (0.1)^2\mathbf{I})$. The four goals are located at $(6, 0)$, $(-6, 0)$, $(0, 6)$, and $(0, -6)$, where the agent can move maximum 1 unit per timestep for each coordinate. The ground-truth reward is given by the difference between successive values of a Gaussian mixture depicted as Fig. 6. The assymetric multi-goal environment has similar settings, except the scale is five times bigger goal of the east side is further located at $(60, 0)$.

- 3D Ant Environment: Let the 2D coordinate representation denotes the orthogonal projection of the position of Ant robot torso. The simulated robot is spawned near the origin. The four goals are located at $(30, 0)$, $(-30, 0)$, $(0, 30)$, and $(0, -30)$, where the agent has to control four legs to reach one of the goals. It requires approximately 150 timesteps for an expert to reach one of the goals from the initial position. A vector of the current position and the robot's joint values represent the state. Since it is hard to train a single expert model to represent the desired multi-modal behavior precisely, we evenly merged 2,000 trajectory samples from 8 uni-modal policies specialized in moving to one of the fixed positions.

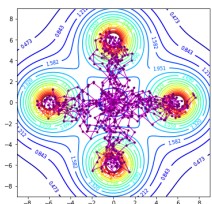 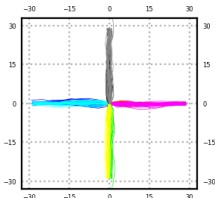

Figure 6: Illustrations of multi-goal environments. From the initial agent position, four goals are symmetrically located at the four cardinal directions. The ground-truth energy function of the 2D environment and the expert trajectory samples for each environment are displayed.

### C.2  TRANSFER LEARNING ENVIRONMENTS

The experiment setting was designed to show that imitation learning for realistic tasks regardless of domain shifts. We trained each IRL network in the target (test) environment. To simply put, our experiment aims to measure the flexibility of reward learning process. We used 1,000 trajectories from the source task as transfer learning data.

- SlopeHopper & SlopeWalker2d: We used the same 3D model from the Hopper-v3 and Walker2d-v3 Mujoco benchmarks. For each task, the ground is tilted with a certain degree in $[1, 15]$. For computing a state vector, the models' height is adjusted to the vertical distance from the slope.

- CrippledAnt: Compared to the original Ant model, we shortened the length of two forelegs to half. We additionally restricted the joint angles of forward legs in the range of $\{.01, .25, .50, .75, 1.0\}$. While the objective of the agent is for running to the right side, the reward can be calculated by the velocity of robot moving along the x-axis.

# D    IMPLEMENTATION DETAILS

## D.1    ALGORITHMS

Algorithm 1 summarizes the overall IRL procedures. The term "REG" in Line 7 of the algorithm refers to the shaping regularization minimizing squared L2-norm $\|\gamma h_{\bar{\psi}}(s') - h(s)\|_2^2$ with the regularization parameter $\lambda \in \mathbb{R}^+$ in order to make the overall algorithm well-defined. For stochastic action distributions, the temperature parameter $\alpha$ has to be multiplied for exact matching between conditional distributions. For ensuring Lipschitz continuity, every critic network is regularized by a gradient penalty (Zhou et al., 2019).

---

**Algorithm 1** Causal adversarial inverse reinforcement learning.

---

1: **Input:** Expert trajectory dataset $\{\tau_E^{(i)}\}_{i=1}^N$, temperature $\alpha$.
2: Initialize policy $\pi_\phi$ and critic functions $D_\varphi$, and $D_{\theta,\psi}$.
3: **for** step $i$ in $\{1, \ldots, N\}$ **do**
4:      Collect $\tau_\pi$ by executing $\pi_\phi$.
5:      $\{(s_t, a_t, \cdot, s'_t)\}_{t=1}^T = \tau_\pi$, $\{(\bar{s}_t, \bar{a}_t, \cdot, \bar{s}'_t)\}_{t=1}^T = \tau_E^{(i)}$
6:      Update $D_\varphi$ using $\frac{1}{T}\sum_{t=1}^T \big[\log(1 - D_\varphi(s_t)) + \log D_\varphi(\bar{s}_t)\big]$ with GP.
7:      Update $D_{\theta,\psi}$ using $\frac{1}{T}\sum_{t=1}^T \big[\log(1 - D_{\theta,\psi}(s_t, a_t, s'_t)) + \log D_{\theta,\psi}(\bar{s}_t, \bar{a}_t, \bar{s}'_t)\big]$ with REG+GP.
8:      Update $\pi_\phi$ with $\alpha \cdot r_\theta(s, a)$ using a maximum entropy policy optimization method.
9: **return** $\pi_\phi, r_\theta$

---

For a policy optimization in the experiments we used PPO implementation train policy by a clipped surrogate objective function as follows:

$$\mathcal{L}_{\pi_{\text{old}}}^{\text{CLIP}}(\phi) = \mathbb{E}_{\pi_{\text{old}}}\left[\min\left(\frac{\pi_\phi(a|s)}{\pi_{\text{old}}(a|s)}A_\alpha^{\pi_{\text{old}}}(s,a), \text{clip}\left(\frac{\pi_\phi(a|s)}{\pi_{\text{old}}(a|s)}, 1-\varepsilon, 1+\varepsilon\right)A_\alpha^{\pi_{\text{old}}}(s,a)\right)\right] \quad (15)$$

where $\varepsilon$ is the clipping range of PPO algorithm, ensuring that the updated policy does not diverge too far from the previous distribution. One thing to be careful in the implementation is that the KL penalty terms are dependent on the policy distribution. Therefore for a PPO algorithm with the KL regularization, the term $A_\alpha^{\pi_{\text{old}}}(s,a)$ can be computed as follows:

$$V^{\pi_{\text{old}}}(s) = \mathbb{E}\left[\sum_{t=0}^\infty \gamma^t(r_\theta(s_t, a_t) - \alpha\log{}^{\pi_\phi(a_t|s_t)}/u)\big|s_0 = s, \pi_{\text{old}}\right], \quad (16)$$

$$Q^{\pi_{\text{old}}}(s,a) = r_\theta(s,a) + \mathbb{E}\left[\sum_{t=1}^\infty \gamma^t(r_\theta(s_t, a_t) - \alpha\log{}^{\pi_\phi(a_t|s_t)}/u)\big|s_0 = s, a_0 = a, \pi_{\text{old}}\right], \quad (17)$$

$$A^{\pi_{\text{old}}}(s,a) = Q^{\pi_{\text{old}}}(s,a) - V^{\pi_{\text{old}}}(s), \qquad \text{(Advantage estimate)} \quad (18)$$

$$A_\alpha^{\pi_{\text{old}}}(s,a) = A^{\pi_{\text{old}}}(s,a) - \alpha. \quad (19)$$

We refer the work of Shi et al. (2019) for the detailed derivation of entropy-regularized policy gradient algorithms.

## D.2    NETWORK ARCHITECTURES AND HYPERPARAMETERS

For all policy and discriminator networks, we use networks with 2-layer MLP with 256-dim layers, and the activation function is ReLU. For input layer, we normalize inputs (state and action vectors) by calculating exponential moving average and variance.

**Details for Policy Networks** For multi-goal environments, the policy function is represented by a Gaussian mixture with four modes. In contrast, a single diagonal Gaussian function is used for other unimodal tasks. For enforcing action bounds, we apply an invertible squashing function (tanh) to the raw action samples and compute the likelihoods of the bounded actions such as `tanh(Normal(`$\mu, \sigma$`))` (Haarnoja et al., 2018). In our implementation, we separated the distribution network into mean network and standard deviation network. The gradient penalty regularization is applied to the value network.

**Details for CAIRL Networks** For CAIRL networks, the reward network consists softplus activation at the final output layer while the potential network has linear activation. Gradient penalty is also used in the both networks. We do not explicitly construct the target potential networks, instead we use the same potential network, but the target potential network does not involved any gradient computation. Also, to discard the learning of terminal state values, $h_{\bar{\psi}}$ is outputs as follows:

$$h_{\bar{\psi}}(s) = \begin{cases} 0 & \text{if } s \text{ is a terminal state not caused by time limits,} \\ \texttt{no\_grad}\big(h_\psi(s)\big) & \text{otherwise} \end{cases}$$

Table 3 shows the hyperparameters of conducted experiments.

Table 3: Hyperparameters.

(a) Shared parameters

| Parameter | Value |
|---|---|
| Optimizer | Adam$(0, 0.99)$ |
| Learning rate (policy) | $1 \cdot 10^{-4}$ |
| Learning rate (discriminator) | $4 \cdot 10^{-4}$ |
| Discount factor ($\gamma$) | 0.99 |
| Clipping Range ($\varepsilon$) | 0.2 |
| Gradient penalty ($\eta$) | $10^{-4}$ |
| Shaping regularization ($\lambda$) | $10^{-4}$ |
| Batch size per rollout | 2048 |
| Policy rollouts | 3 |
| Discriminator rollouts | 3 |
| Gradient steps per iteration | 15 |

(b) Environment specific parameters

| Environment | # of threads | Temp. ($\alpha$) |
|---|---|---|
| Multi-goal (Point) | 1 | 1 |
| Multi-goal (Ant) | 16 | $10^{-2}$ |
| Humanoid-v3 | 8 | $10^{-3}$ |
| Walker2d-v3 | 3 | $10^{-3}$ |
| Others | 1 | $10^{-2}$ |

# E  ADDITIONAL EXPERIMENTAL RESULTS

## E.1  ADDITIONAL BENCHMARK RESULTS

Table 4: The evaluation on imitation learning benchmark control tasks presented over 5 different random seeds with 4 and 100 expert trajectory data.

| Method | Environment | | | | |
|---|---|---|---|---|---|
| | Hopper-v3 | HalfCheetah-v3 | Walker2d-v3 | Ant-v3 | Humanoid-v3 |
| Expert | $3688.9_{\pm 358}$ | $3580.3_{\pm 135}$ | $5297.0_{\pm 696}$ | $4030.8_{\pm 839}$ | $9049.0_{\pm 3114}$ |
| GAIL | $3227.6_{\pm 403}$ | $\mathbf{3431.9_{\pm 251}}$ | $2369.8_{\pm 507}$ | $\mathbf{4090.8_{\pm 252}}$ | $\mathbf{8816.5_{\pm 504}}$ |
| AIRL | $\mathbf{3229.8_{\pm 341}}$ | $3127.6_{\pm 296}$ | $\mathbf{3326.7_{\pm 1231}}$ | $3956.5_{\pm 302}$ | $538.8_{\pm 580}$ |
| FAIRL | $374.1_{\pm 300}$ | $1147.4_{\pm 330}$ | $281.4_{\pm 116}$ | $1501.3_{\pm 371}$ | $351.9_{\pm 30}$ |
| CAIRL | $\mathbf{3565.4_{\pm 233}}$ | $\mathbf{3163.5_{\pm 297}}$ | $\mathbf{5089.6_{\pm 339}}$ | $\mathbf{3990.9_{\pm 254}}$ | $\mathbf{9644.6_{\pm 385}}$ |
| GAIL | $\mathbf{3513.1_{\pm 273}}$ | $3493.9_{\pm 279}$ | $4392.4_{\pm 391}$ | $4093.2_{\pm 258}$ | $9199.4_{\pm 443}$ |
| AIRL | $3349.0_{\pm 318}$ | $3482.6_{\pm 202}$ | $\mathbf{5165.7_{\pm 205}}$ | $4027.0_{\pm 282}$ | $8532.5_{\pm 720}$ |
| FAIRL | $236.2_{\pm 10}$ | $744.2_{\pm 249}$ | $281.4_{\pm 116}$ | $2106.0_{\pm 372}$ | $375.2_{\pm 22}$ |
| CAIRL | $\mathbf{3571.6_{\pm 309}}$ | $3246.4_{\pm 244}$ | $\mathbf{5268.1_{\pm 194}}$ | $3979.6_{\pm 296}$ | $\mathbf{9469.5_{\pm 374}}$ |

## E.2  VISUALIZATION OF TRAJECTORIES GENERATED BY IMITATION LEARNING AGENTS

Fig. 7 shows the trajectories from the trained models in the 3D environment with different seeds. Firstly, the result suggest that CAIRL recovers multi-modal policies which is equivalent to the property MaxEnt IRL. At the same time, CAIRL is fundamentally advanced algorithm compared to MaxEnt IRL algorithms because it can be applied various large domain without knowing dynamics by by enlarging number of parameters in neural networks.

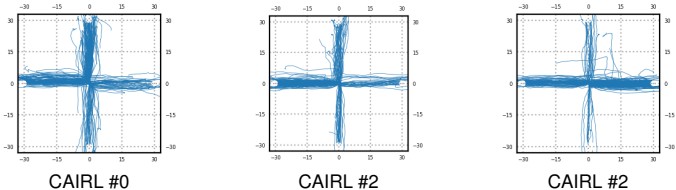

Figure 7: Multi-goal Ant trajectories with different random seeds

### E.3 ABLATION STUDY ON HYPERPARAMETERS

We provide experiments on ablation study on three controllable hyperparameters in Fig. 8.

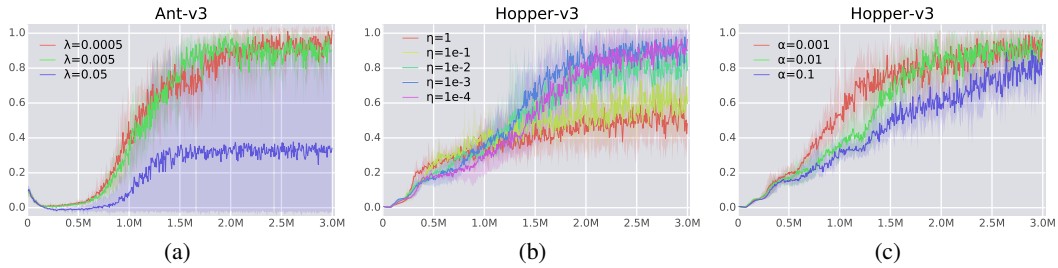

Figure 8: Experiments on hyperparameters of CAIRL. (a) If the shaping regularization $\lambda$ is excessively high, learning of the potential-based shaping is disabled, and the reward function shows unstable behavior. (b) The gradient penalty is applied for ensuring convergence of discriminators. The results suggest that sufficiently low gradient penalty is preferred for achieving desired performance. (c) Compared to the other hyperparameters, CAIRL is robust with the temperature $\alpha$.

### E.4 VISUALIZATION OF CRIPPLEDANT LOCOMOTION

We provide the visualization of generated samples from a trained agent trained by CAIRL algorithm in Fig. 9. Even though dynamics of the environment is considerably misaligned, the algorithm still can teach agent appropriate algorithm to teach an agent to go desired direction. We believe that this work also provided effective transfer learning algorithm for sequential decision problems.

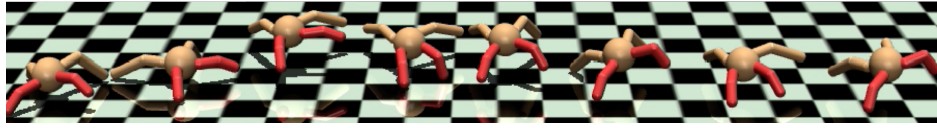

Figure 9: Visualization of trained agent on CrippledAnt environment. The model maximizes the movement speed toward the right side, without showing biased locomotion such as moving to the forward or to the backward. Also, the orientation of the torso is preserved throughout the episode, implying that the algorithm is capable of imitating experts even under variation in the dynamics.

### E.5 VISUALIZATION OF AIL REWARD FUNCTIONS

We provide the visualization of CAIRL and AIRL rewards in the 2D multi-goal environment. To effectively represent the two functions that takes the argument $(s, a) \in \mathcal{S} \times \mathcal{A}$ into the space $\mathcal{S}$, we calculated a two-dimensional vector representation of reward for each state by averaging all possible action in $\mathcal{A}$, i.e., $\mathbf{v}_i = \frac{1}{|\mathcal{A}|} \sum_{a \in \mathcal{A}} [r(s, a) \cdot a_i], i \in \{0, 1\}$. As a results, we plotted contour maps by the relative value of rewards and corresponding vector fields of reward function for a state grid, which is shown in Figs. 10 and 11. Apparently, it can be observed that the CAIRL reward function is much more analogous to the MaxEnt likelihood depicted by Haarnoja et al. (2017). More importantly, CAIRL is prominently showing that our reward modeling provides more informative reward that approximately advise the best direction to reach one of the goals for each state.

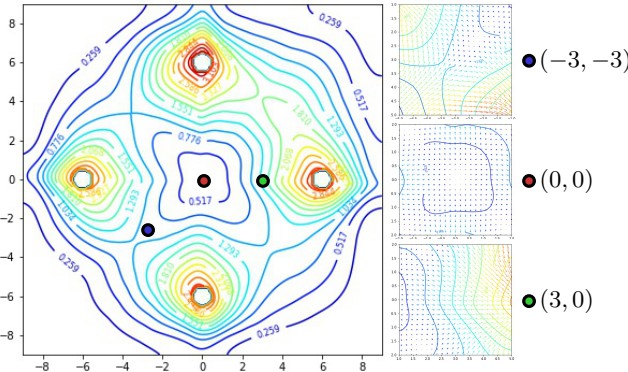

Figure 10: Illustration of the CAIRL reward function in the 2D multi-goal environment. Left: recovered reward function which is averaged over action. The goals are located in the four cardinal direction (6,0), (-6,0), (0,-6), (0,6). Right: visualization of each vector field of reward function and the corresponding local contour map.

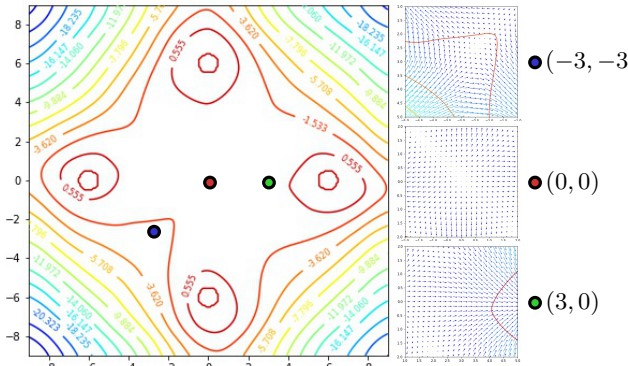

Figure 11: Illustration of the AIRL reward function.

