# OpenReview forum: "Unbiased Learning with State-Conditioned Rewards in Adversarial Imitation Learning"
_ICLR.cc/2021/Conference — Reject_

### Official Review · AnonReviewer1 · 2020-10-20
**Some interesting ideas, but questionable algorithmic choices and flawed evaluation**

**Rating:** 4
**Confidence:** 5

**Review:**

Summary
========
The paper proposes several modifications to adversarial inverse reinforcement learning (AIRL) that aim to improve transfer to different environments, to address termination bias, and to learn more multimodal policies. The most notable change is probably the introduction of an additional discriminator, $D_\phi(s)$, that is trained on the state marginal. By adding the logits of the state-discriminator to the logits of the state-action(-nextState)-discriminator, $D_{\theta,\psi}(s,a,s')$, the latter supposedly estimates the expert's advantage function better. Apart from adding a state-discriminator (1), I could identify the following modifications to AIRL: (2) adding a constant offset to the optimization problem given by the log-density of a uniform action distribution, (3) enforcing positive rewards through a soft-plus, (4) disregarding the gradient of the next states Value $h_{\bar{\psi}}(s')$ when updating the discriminator and (5) penalizing the temporal value differences $||\gamma h_{\bar{\psi}}(s') - h(\psi)||_{2}^2$ via regularization.
The resulting algorithm, CAIRL, is evaluated (a) on a 2D and 3D toy problem for testing the multi-modality of the learned policy (b) on MuJoCo locomotion tasks to evaluate imitation learning performance, and (c) on MuJoCo tasks with modified dynamics to test transferability. The transfer setting is different from the setting used by AIRL because the evaluation learns the policy and reward function directly in the test domain and it is not assumed that the original domain can be simulated. Hence, the evaluation tests the robustness of the imitation learning rather than the robustness of the reward function. Especially at the transfer experiments, CAIRL performed better than AIRL.

Strong points
===========
- The paper is well-written, tables and figures are helpful
- Evaluations on different environments with respect to different properties (multimodality, imitation learning, transfer learning)
- The paper comes with source code

Weak Points
===========
- (I) I don't see the motivation for most of the modifications to AIRL. I do not see how they are supposed to improve multimodality or transferability
- (II) The evaluations seem to be termination-biased by using positive rewards for CAIRL. Hence, the results provide little value
- (III) the changes seem to be incremental at best (and quite possibly detrimental, esp. (3-5) )
- (IV) Provided source code only contains demonstrations for the "Hopper" imitation learning experiment. There is no straightforward way to reproduce the other evaluations

Recommendation
===========
I recommend rejection because there seems to be little logic behind the proposed modification. I don't see the theoretical basis for the claimed benefits and the empirical analysis is possibly dominated by reward bias that was introduced by restricting CAIRL to positive rewards.

Supporting Arguments
==================
My recommendation for rejecting is primarily due to the weak point (I) and secondarily due to the weak point (II). Hence, I will now focus on these points.

I
---

The motivations for most of the motivations are either missing or dubious.
- __Adding a state discriminator (1):__ The paper claims that the additional discriminator improves transferability (based on Eq. 9). AIRL minimizes the reverse KL of the state-action distributions using a reward function that for an optimal discriminator is given by $r^\star(s,a) \approx \log \frac{\rho_{E}(s)}{\rho_{\pi}(s)} \log \pi_{E}(a|s)$. CAIRL minimizes the KL between the policies using a reward function that for an optimal discriminator is given by $r^\star(s,a) \approx \log \pi_{E}(a|s)$. Intuitively, the term $\log \frac{\rho_{E}(s)}{\rho_{\pi}(s)}$ that was dropped by CAIRL, seems to be very useful by pushing the policy towards the demonstrated states and away from those states that are not encountered by the expert. This additional reward to bring the policy close to demonstrated states seems to be a major advantage of AIRL compared to direct behavioral cloning that is known to suffer from covariate shift due to compounding policy errors [1]. It is not clear to me why removing this term improves imitation learning when learning under different dynamics. Based on Eq. 9 the expected unshaped cost corresponds to the KL $D_{\text{KL}}\left[ p_{\pi_\phi}(a,s'|s) || p_{E}(a,s'|s) \right]$ and thus minimizes the divergence between these distributions that are obtained under different dynamics. However, wouldn't AIRL analogously be penalized based on the KL $D_{\text{KL}}\left[ p_{\pi_\phi}(a,s',s) || p_{E}(a,s',s) \right]$? Why is a policy that puts less emphasis on matching the demonstrated states expected to perform better in a different environment? I would argue that it is more likely that good states remain good under different dynamics than that good actions remain good, but of course, it always depends on the concrete setting.

- __Adding a constant term to the objective (2)__: I am fine with modification (2) since I see how the constant offset of $u = \log \pi_{\text{uniform}}$ can tackle termination bias for CAIRL. However, the paper motivates this modification as a general, simpler way of tackling termination bias compared to the approach by Kostrikov et al. (2018), which is highly misleading. I think Section 4.2. needs to be rewritten.

- __Enforcing positive rewards (3)__: This modification is in my opinion a crude hack that invalidates much of the evaluation based on a dubious motivation by arguing that the expected reward is positive. First of all, the fact that the expectation of a function is positive does not imply that the function itself is positive. Furthermore, the $\ge$ sign on page 6 is wrong; it should be $E_{\pi} \left[ \log \pi_E \right] \le E_{\pi} \left[ \log \pi \right]$ and not the other way around. On a side note, there is also a typo in that equation, because I suppose the log should be in front of the fraction.

- __Disregarding the gradient of the next state's value (4)__: The paper argues that this is analogous to target networks in deep RL. However, (C)AIRL is not training the value net by minimizing the mean squared TD error but minimizes cross-entropy loss based on a special discriminator architecture, so I do not see the connection here.

- __Penalizing the temporal value differences (5)__: The regularizer would make sense to me if it was penalizing the TD error. However, I don't see the reason for penalizing $||\gamma h_{\bar{\psi}}(s') - h(\psi)||_{2}^2$. The paper just argues that this regularizer makes "the overall algorithm well-defined", without providing any reasoning.

II
----
Modification (3) adds a reward bias. Enforcing that all state-action pairs produce positive reward introduces survival bias. Entering unhealthy states in the locomotion tasks would enter an absorbing state with a hard-coded reward of 0. Hence, the agent will avoid entering such states. The survival bias might be a major reason, why CAIRL performs better than AIRL when training in a modified environment.
I also think that evaluation not fair by not addressing reward bias for AIRL. CAIRL makes use of information about environment resets due to unhealthy states by assigning a value of zero to these states. AIRL could also make use of this information by addressing termination bias as proposed by Kostrikov et al. (2018).



Questions
----------
- At the bottom of page 4 you argue that compared to AIRL, the CAIRL reward can be "applied for training arbitrary policies since the agent-specific occupancy measure $\rho_\pi(s,a)$ is detached". Why can't the AIRL reward be used for training arbitrary policies?

- Why is CAIRL assumed to be better at learning multi-modal policies? The connection between CAIRL and MaxEnt-IRL is based on the AIRL derivations, so I do not see the reasoning here.

Additional Feedback
--------------------
I think more details should be provided for the multimodal experiment. Can you plot the demonstrations as well?

References
----------
[1] S. Ross and D. Bagnell.  Efficient reductions for imitation learning.  AISTATS. 2010.

---

> ### Author Response · Authors · 2020-11-20
> **Response to Official Review #1**
>
> * **Motivation**. We think this is the most fundamental criticism among reviews. We have decided to resolve this point in the new version. In short, we examined the claim starting from an argument that the AIL algorithms do not correctly form an EBM, which was initially raised by Liu et al. (2020).
> * **There is no straightforward way to reproduce other evaluations.** We believe we have provided a sufficient amount of information about the experiments, including: algorithms, environments, how data is used, and the actual data that we have used. As we did not intend to make the reviewers confused, we include additional demonstration data of Ant-v3 and Walker2d-v3. We planned to release the entire trajectory data after we are certain about the situation.
> * **Adding a state discriminator:** We would like to make it clear that reward functions derived by our method, which is an element of $\mathfrak{R}$, do not entirely move away useful state transition that happens in the future as concerned. Because, 1) following the appropriate action will eventually minimize the discrepancy between the transition states by the MDP dynamics, and 2) in the subsequent RL, the averaged return can empirically correct the trajectory distribution. In fact, in particular case of $\gamma\approx 1$, since the CAIRL reward recovers an state-conditioned reward function, the following relationship of the KL divergence of the conditional trajectory of expected return can be obtained: $\mathbb{E}[\sum_{t=0}^\infty \mathrm{D}_\mathrm{KL}[\pi(\cdot|s_t)\Vert\pi_E(\cdot|s_t)]|\pi]=\mathrm{D}_\mathrm{KL}\bigl[\Pr(\tau|s_0 = s,\pi)\Vert\Pr(\tau|s_0 = s,\pi_E)\bigr].$
> * "Wouldn't AIRL analogously be penalized based on the KL divergence?" Since this argument does not hold, previous AIL works alternatively invented a convenient term called "generalized" divergence, a pseudo formulation of the probabilistic metrics using occupancies. In particular, if AIRL is applied in the transfer learning tasks, we assume the formation would be $\mathrm{D}_\mathrm{KL}\bigl[\rho_\pi(s,a,s^\prime) \big\Vert \rho_E(s,a,s^\prime)\bigr]$ by additionally defining state-action-state occupancy measures. Nevertheless, keep in mind that the use of the custom function $\rho_\pi(s,a,s^\prime)$ is not thoroughly analyzed, so we are not sure about the outcome when it is represented as a cumulative sum. In contrast, $\mathrm{D}_\mathrm{KL}\bigl[p_\pi(s^\prime|s,a) \Vert p_E(s^\prime|s,a)\bigr]$ is the actual probabilistic metric that we can find what would happen when learning them as an energy-based model.
> * **Adding a constant term to the objective:** We appreciate the suggestion and have rewritten Sec. 4.3. We would like to add a comment that the word "simpler" was used to emphasize that our method is actually practical when implementing each algorithm.
> * **Enforcing positive rewards:** We agree with this point; we deleted the equation in the revision acknowledging that it is misleading, and it is not the core part. As the terminal states always get the highest entropy bonuses (which is equal to 0 in the KL regularization), the positivity constraint does not exploit awareness of terminal states. CAIRL formulation allows arbitrary reward functions as long as they reconstruct the term $\log(\pi_E(a|s)/u)$. Since the algorithm solves both types of tasks that require survival and termination behaviors, we think the choice of activation function can be allowed as one of the settings.
> * **Disregarding the gradient of the next state's value:** We understand your concern: CAIRL is not performing the exact TD learning by minimizing the mean squared TD error. However, what we are showing is that the *outcomes* after TD learning and CAIRL learning are equivalent. For example, we can choose to minimize TD error with minimization of an arbitrary $p$-nom, and we can confidently say TD learning will be converged, no matter of $p$.
> * "Hard-coded reward of 0": To the best of our knowledge, assuming 0 rewards *after* terminal states is not hard-coding at all. In fact, it is the most reasonable setting to define the sum of rewards as a return, and is implicitly used in almost all finite problems across the RL/IRL domain. In particular, what CAIRL is not doing are 1) adding synthetic transitions 2) manipulating the actual return; therefore, CARIL does not use information about environment resets. In other words, the reward of 0 after termination is not assigned, it is naturally drawn by the RL formalism.
> * "Why can't the AIRL reward be used for training arbitrary policies? & "Why is CAIRL assumed to be better at learning multi-modal policies?" We agree that this parts are little bit obscure. Therefore, we have decided to fix this point by highlighting CAIRL recovers an energy-based model.
> * Plotting the demonstrations: This information had been provided in the appendix of the original submission. The main text of our revision briefly mention this information in the main text of our revision.

---

> > ### Comment · AnonReviewer1 · 2020-11-20
> > **Further Remarks**
> >
> > __Regarding Reproducibility__: "We planned to release the entire trajectory data after we are certain about the situation." Certain about which "situation"? I do not see the reason for not sharing the trajectory data and the code for running these experiments already now.
> >
> > __Regarding the motivation for the state discriminator__:
> > - I can't follow your argumentation regarding discounting. From what I understand, CAIRL would only minimize the policy-KL $\mathbb{E} \left[ \sum_{t=0^T} D_{\text{KL}}  \left[ \pi(\cdot|s_t)||\pi_{E}(\cdot|s_t) \right] | \pi \right]$ under the assumption that CAIRL is applied to the same environment that was used by the expert. However, in that setting, AIRL (under the same strong assumption that the optimal discriminator is learned) would also solve the task for any discount factor, because we get $p_{\pi}(s) = p_{E}(s)$. For me it seems like both approaches would produce the same policy under strong optimality assumptions, whereas AIRL produces slightly more similar trajectories when they can't be exactly matched, by assigning additional penalty for states that the learned policy visits more often than the expert.
> >
> > - I'm also not following the argument about generalized divergences, "previous AIL works alternatively invented a convenient term called "generalized" divergence". Note that generalized divergences is not at all an invention by AIL works but a well-established extension of divergences to non-normalized measures. It is also not clear to me why AIRL would minimize  a generalized divergence.
> >
> > __Additional Remark on transfer guarantees__:
> > I don't find a convincing proof that the reward function learned by CAIRL minimizes the trajectory KL when it was learned under different system dynamics.
> > - Equation 9 considers the agent's dynamics (the expectation is with respect to $\mathcal{P}$), so I don't see what it tells us about the expert's MDP. Indeed, maximizing the function $f(s,a,s')$ in the expert's MDP does not seem to minimize an expected KL since we would get
> > $$ \underset{\pi}{\max} \int_{s} p_{E}^{\pi}(s)  \int_{a,s'}  \pi(a|s) \mathcal{P}_E(s'|s,a) \log \frac{ \mathcal{P}_E(s'|s,a) \pi_E(a|s) }{ \mathcal{P}(s'|s,a) \pi(a|s)}.$$
> > Note that the agent's policy $\pi$ is paired with the expert's dynamics for computing the expectation, but it is paired with the agent's dynamics inside the expectation.
> >
> > - Furthermore, the paper claims that maximizing $r(s,a)$ "promotes the identical effect" compared to maximizing $f(s,a,s')$. This statement is wrong for the transfer setting because $r(s,a)$ is an unshaped version of $f(s,a,s')$ only for the system dynamics that were used during training.
> >
> > __Regarding positive rewards__:
> > While it is good that the revision does no longer contain the previous motivation which was flawed for several reasons, the current submission does not seem to discuss this hack at all anymore. You argue that "CAIRL formulation allows arbitrary reward functions as long as they reconstruct the term $\log \frac{\pi_{E}(a|s)}{u}$", but this is violated by restricting to positive rewards. So I see only reasons against enforcing positive rewards, but no justification for doing it. Of course, there is empirical justification, since positive rewards are survival biased and thus  help for most experiments.
> >
> > __Regarding Disregarding the gradient of the next state's value__:
> > I'm not satisfied with your reply. The fact that the optimal classifier would (under extremely unrealistic assumptions) result to zero TD-error does not mean that the loss function treats $r(s,a) + \Phi(s')$ as target for $\Phi(s)$. Can you provide a justification for the CAIRL setting that is not based on analogies to a quite different problem setting?
> >
> > __Regarding Hard-coding of zero rewards for absorbing states__:
> > Note that RL and IRL theory is based on observing all states of the MDP, including absorbing states. As Kostrikov et al. pointed out, many common RL frameworks don't provide absorbing states to the learner and thereby implicitly assume a reward of zero. This is fine in the RL setting, because the practitioner can take this into account when designing their reward function. However, in IRL this results in reward bias, since the learning algorithm does not know about this implicit assumption. Hence, Kostrikov et al. ensured that also absorbing states are presented to the discriminator, such that their reward can be learned by the algorithm.  Note that this does not add "synthetic" transitions, but real transitions that are present in the actual MDP but hidden by common implementations (e.g. baselines). By not allowing the algorithm to learn the reward for absorbing states their theory breaks. Also, my point of "II" was that CAIRL is explicitly survival-biased by enforcing positive rewards. I think for a fair comparisons all implementations should not be reward-biased in any way.

---

> > > ### Author Response · Authors · 2020-11-25
> > > **Response to Further Remarks (1/2)**
> > >
> > > **Regarding Reproducibility:** In response to the reproducibility concern, we uploaded most of the trajectories used for the benchmark experiments (some are omitted due to the maximum file size 100MB). We also have updated the source code for including ablation study features.
> > >
> > > **Regarding the motivation for the state discriminator:**
> > > * From the architecture, it can be thought that AIRL would more sensitively react to state discrepancies. However, we would like to make it clear that in theory the local condition of a discrepancy between $\rho_\pi(s_t)$ and $\rho_E(s_t)$ does not affect the learning of $\pi_\phi(\cdot|s_t)$ for the current time-step $t$ in the standard RL algorithms (i.e., policy gradient and actor-critic variants), because the derivative of function regarding state will be eliminated when computing the gradient w.r.t. $\phi$. Hence, we need to examine additionally providing the information of discrepancies between  for the subsequent time-steps, namely $\mathrm{D}_f\bigl(\rho_\pi(s_{t+1})\big\Vert \rho_E(s_{t+1})\bigr), \dots$, would be beneficial to teaching behavior of $\pi(\cdot|s_t)$.
> > >
> > >   A quite lengthy, but general derivation of  CAIRL would be as follows: $\mathbb{E}\bigl[\sum_{t=0}^{T-1} \gamma^t \log\bigl(\frac{P_E(s_{t+1}|a_t,s_t)}{P(s_{t+1}|a_t,s_t)}\bigr)\mathrm{D}_\mathrm{KL}\bigl[\pi(\cdot|s_t)\Vert \pi_E(\cdot|s_t)]\big|\pi\bigr]=\mathbb{E}\bigl[\sum_{t=0}^{T-1} \gamma^t\mathrm{D}_\mathrm{KL}[p_\pi(s^\prime_t,a_t|s_t)\Vert p_E(s^\prime_t,a_t|s_t)]\big|\pi\bigr]$---(a).
> > >
> > >   In this paper, we claimed the MaxEnt RL of $\pi(a|s)$ would be trained with the target distribution, which is located at a projection between $\frac{\exp\{\log p_E(s^\prime,a|s)\}}{\sum_{\bar{s}^\prime,\bar{a}} \exp\{\log p_E(s^\prime,a^\prime|s)\}} = p_E(s^\prime,a|s)$ to $\frac{\exp\{\log \Pr(\tau=\tau^\prime,a_0=a|s_0=s,\pi_E)\}}{\sum_{\bar{\tau}} \exp\{\log\Pr(\tau=\bar{\tau}|s_0=s,\pi_E)\}} = \Pr(\tau=\tau^\prime,a_0=a|s_0=s,\pi_E)$ depending on the value of $\gamma$. Note that the two ways of marginalization $\sum_{s^\prime} p_E(s^\prime,a|s)$ and $\sum_{\tau^\prime}\Pr(\tau=\tau^\prime,a_0=a|s_0=s,\pi_E)$ equally yield the ground-truth $\pi_E(a|s)$ when the distributions of trajectories can be matched. When they cannot be matched, CAIRL will alternatively choose to learn the best policy for the current state that can minimize $\sum_{t=0}^{T-1}\gamma^{t}\cdot\mathbb{E}_{s_t\sim\tau_\pi}[\mathrm{D}_\mathrm{KL}[p_\pi(s^\prime_t,a_t|s_t)\Vert p_E(s^\prime_t,a_t|s_t)]]$  (as the expectation operator is linear, we can interchange the expectation and the summation in Eq. (a)). In contrast, when the MaxEnt RL is applied on the vanilla AIRL reward function, we do not believe that each $\pi(a|s)$ will be projected to $\pi_E(a|s)$ in the ideal (imitation learning) case, for fixed discriminator in mentioned property. We empirically proved our point that providing the state densities does not help domain adaptation in the general (transfer learning) case.
> > > * It is likely that the notion of generalized divergences had been used in other imitation learning algorithms before the GAIL algorithm (Ho & Ermon, 2016); we would like to correct this response if it is true. Yet, what we wanted to emphasize here is that, the one-to-one correspondence between occupancy measures $\rho_\pi(s,a)$ and policy distribution $\pi(a|s)$ (Theorem 2 of Syed et al., 2008), does not explain what would happen learning with occupancy measures in different or finite horizon MDPs. For an answer to AIRL formulation, since GAIL discriminator  will be converged to $D(s,a)=\frac{\rho_E(s,a)}{\rho_E(s,a)+\rho_\pi(s,a)}$ and AIRL has the architecture $\frac{\exp(f(s,a))}{\exp(f(s,a))+\pi(a|s)}$, the reward function of AIRL will be converged to $f(s,a)=\log\frac{\rho_E(s,a)}{\rho_\pi(s)}$.  It can be derived that AIRL minimize as follows: $\mathbb{E}_\pi\bigl[f(s,a)-\log\pi(a|s)\bigr]=\int\int\rho_\pi(s,a)\log\frac{\rho_E(s,a)}{\rho_\pi(s,a)}\ \mathrm{d} s\mathrm{d} a \triangleq -\mathrm{D}_\mathrm{KL}\bigl[\rho_\pi(s,a)\big\Vert\rho_E(s,a)\bigr]$. Therefore, we can make an expression that AIRL minimizes the generalized reverse KL divergence.

---

> > > > ### Author Response · Authors · 2020-11-25
> > > > **Response to Further Remarks (2/2)**
> > > >
> > > > **Additional Remark on transfer guarantees.**
> > > > * Our design was that the transfer learning would happen in the RL process of the IRL framework, in the agent's (current) MDP, using the IRL reward function. Therefore, in our settings, the expert's MDP cannot be attained and simulated; the difference between dynamics can only be approximated by the discriminator network using the expert trajectory data. Therefore, maximizing the function $f(s,a,s^\prime)$ will be performed in the agent's MDP, and we get $\max\limits_\pi\int_S\rho_\pi(s)\int_A\int_S \pi_E(a|s)\mathcal{P}_E(s^\prime|s,a)\log\frac{\pi_E(a|s)\mathcal{P}_E(s^\prime|s,a)}{\pi(a|s)\mathcal{P}(s^\prime|s,a)}\ \mathrm{d}s^\prime\ \mathrm{d}a\ \mathrm{d}s$.
> > > > * Recall that the theorem of reward shaping theory holds regardless of the transition distribution. Since the MDP in which learning is happening  is consistent during the RL phase, the reward function $r_\theta(s,a)$ promotes the identical effect with $f(s,a,s^\prime)$ in the agent's (current) MDP.
> > > >
> > > > **Regarding positive rewards:** We understand your concern about positive rewards. To answer this point, we conduced ablation experiments and  updated Fig. 4, which now includes the results with a linear activation function. Although soft plus activation helped the overall stability, but CAIRL with linear activation also achieved competitive scores; and outperformed the softplus version in one task (Walker2d-v3).
> > > >
> > > > **Regarding Disregarding the gradient of the next state's value:** According to Geist et al., (2019), there is only one fixed point of the optimal value function in regularized MDPs (unlike traditional value learning of RL), and the analogy between two learning methods is drawn by observing the equivalent convergence point. We acknowledge this can be an extreme condition.
> > > >
> > > > If you are seeking for the empirical justification for disregarding the gradient, we can explain in this way. Suppose  we measure the difference between state $\Vert s_{t+1} - s_t\Vert = \Delta$. If average $\Delta$ is very small and $\gamma$ is close to 1, then the learning of $\gamma h_\psi(s_{t+1}) - h_\psi(s_t)$ does not efficient learn the desired quantities as expected and $\gamma h_\psi(s_{t+1})$ causes negative affect of learning $h_\psi(s_t)$ in practice.
> > > >
> > > > **Regarding Hard-coding of zero rewards for absorbing states:**  We started with a question: "what would an expert MaxEnt RL policy exactly represent in absorbing states?", and we derived an obvious answer of a uniform distribution. We are fully aware that our answer disputes the major concept of Kostrikov et al. However, we want to stress that the paper builds on a particular case, where the reward learning for absorbing states is not required using a specific regularization scheme in MaxEnt frameworks.

---

> > > > > ### Comment · AnonReviewer1 · 2020-11-25
> > > > > **Remaing Points**
> > > > >
> > > > > __Additional Remark on transfer guarantees__
> > > > > I think that I do understand your setting of training a policy in a different environment. However, the paper claims several times that the learned _reward function_ transfers to different environments. To test whether a learned reward function transfers to a different environment, one would need to maximize that (fixed) reward function in the new environment and inspect the resulting behavior. When we do this with the CAIRL reward, it will no longer minimize the KL, because the expectation in the Max-RL objective would now be computed with respect to the new dynamics.
> > > > >
> > > > > Also, maximizing $r(s,a)$ would no longer result in the same policy as maximizing $f(s,a,s')$ since the relation $f(s,a,s') = r(s,a) + \gamma h(s') - h(s)$ would no longer hold, as the next-states $s'$ are now drawn from a different distribution.
> > > > >
> > > > > __Regarding positive rewards__
> > > > > Thanks for providing ablations without this hack.
> > > > >
> > > > > __Regarding Disregarding the gradient of the next state's value__
> > > > > It might be the case that $h(s)$ will still converge to the optimal value function, but I still do not see any proof for it.
> > > > >
> > > > > __Regarding  Hard-coding of zero rewards for absorbing states__
> > > > > I actually don't see how you dispute the major concept of Kostrikov et al. From what I can tell you provide a different solution for addressing reward bias that is very specific to CAIRL. Still, this does not justify to address reward bias for CAIRL but not for the competitors during the evaluations.

---

> > > > ### Comment · AnonReviewer1 · 2020-11-25
> > > > **Regarding the motivation for the state discriminator**
> > > >
> > > > First of all thanks a lot for uploading the additional data.
> > > >
> > > > Regarding the motivation for the state discriminator:
> > > > Unfortunately your "quite lengthy, but general derivation of CAIRL" is not very helpful, since the Equation does not make any sense to me.  Why is the log-density ratio $\log\frac{p_E(s_{t+1}|a_t, s_t)}{p(s_{t+1}|a_t, s_t)}$ multiplied with the policy-KL? I'm also not certain about the setting you are considering here. Which dynamics are used for computing the expectations?
> > > >
> > > > I also don't see sufficient justification for your claim that the AIRL policy would not approximate the expert policy in the ideal imitation learning case. Also the Appendix B.2. seems to provide merely claims and little proof.
> > > > From what I can tell:
> > > > 1) The logits $\sigma(s,a,s')$ of the optimal discriminator approximates the density ratio of the respective data distribution that were used for drawing positive / negative samples, that is $\sigma(s,a,s') = \log \frac{p_{E}(s,a,s')}{p^{\pi}(s,a,s')}$
> > > > 2) By parameterizing $\sigma(s,a,s') = f(s,a,s') - \log \pi(a|s) = \log \frac{p_{E}(s,a,s')}{p^{\pi}(s,a,s')}$, we thus get $f(s,a,s') = \log \frac{p_{E}(s'|s,a)}{p^{\pi}(s'|s,a)} + \log \frac{p_{E}(s)}{p^{\pi}(s)} + \log \pi_{E}(a|s)$
> > > > 3) Hence the MaxEnt-RL objective of AIRL $\underset{\pi}{\max} \int_{s,a,s'} p^{\pi}(s,a,s') f(s,a,s') ds da ds' + H(\pi)$ minimizes $D_{\text{KL}}(\pi||\pi_{E})$ in the idea case where we have $p_{E}(s'|s,a) = p^{\pi}(s'|s,a)$ and $p_{E}(s) = p^{\pi}(s)$, and in the more general case it minimizes $D_{\text{KL}}(p^{\pi}(s,a,s')||p_{E}(s,a,s'))$.
> > > >
> > > > What do I miss? Why should minimizing $D_{\text{KL}}(p^{\pi}(a,s'|s)||p_{E}(a,s'|s))$ be preferred?

---

### Official Review · AnonReviewer4 · 2020-10-28
**A mostly sound work though with somehow poorly justified design choices**

**Rating:** 4
**Confidence:** 5

**Review:**

The line of reasoning and analysis followed in the paper is mostly sound. The paper claims that the use of (state-action) occupancy measure make IL and IRL methods brittle due to the high variance of these measures and their inability to transfer to other domains. These two claims are neither properly defined and grounded in the literature, nor are they isolated experimentally. It is important to show clearly that a) these are problems, b) hitherto unaddressed in the research landscape, and that the proposed methods and techniques address these problems specifically.

The paper also claims to address  the reward bias in imitation learning due to mis-specified termination cases. Since this claim is not tied to the previous claim, there should be clear experiments showing the advantage of proposed methods addressing each claim in isolation. This would show a) whether either of the proposed solutions can be used alone with benefits, and b) what is the individual impact of each contribution. Besides the solution claiming to address the reward bias needs to be directly compared against the method introduced in Kostrikov2019 (cited in paper). On another note, the method addressing the bias seem to suffer from a logic flaw. The paper states that in a “self-looping” terminal state, the expert policy is equivalent to a the uniform action distribution, and designs the reward (unrelated yet to reward shaping strictly-speaking) accordingly (0 reward when the expert coincides in action distribution with the uniform one over actions). Since, according to Figure 1, such a KL penalty seem to be applied at every transition, it is not difficult to imagine scenarios (highly multi-modal expert, expert in exploratory phase, etc.) where the designed reward will be zero in non-terminal state due to a close-to-uniform behavior displayed by the expert. The designed reward will then fatally fail to allow the policy to imitate such expert properly.

At the end of page 4, the paper states: “Compared to the AIRL, […] a reward function r [proposed in the paper] can be universally applied for training arbitrary policies since the agent-specific occupancy measure […] is detached.”. In the absence of empirical or theoretical results showcasing the flaws of learning a metric such as the one in AIRL, the claim remains unclear and unjustified. It must be made clear why the reward formulated by AIRL is flawed. Using results from the “state-only” adversarial imitation learning literature might contribute positively to such a ustification and will echo the first claim, however such results are not mentioned nor cited in the paper. Besides, since AIRL uses state-action occupancy measures in its reward design while the proposed reward involves the policy’s action probability directly, the proposed approach is close in essence to behavioral cloning. It differs from BC as it also involves reward shaping, as introduced in Ng1999 (cited in paper). The similarities in reward formulation appears clearly in the table from Ghasemipour2020 (cited in paper), from which Table 1 is heavily inspired. The BC line should be kept to highlight how similar the formulations are. Moreover, while the introduction of potential-based shaping is justified in AIRL, it is not motivated here, and it remains unclear why it was introduced.

It should be made clearer in the text that the method builds on AIRL, considering it involves of the functional elements introduced in AIRL (shaping, EBM formulation, MaxEnt IRL, derivations, etc.).

About the experiments that are presented: the results of CAIRL are encouraging. As said earlier, it is difficult to disentangle the contribution of each component without ablation of the introduced techniques. It would also be interesting to evaluate the proposed IRL method on IRL tasks (reward recovery, as in AIRL), not only on IL tasks (perform well in environment).
The transfer learning experiments show high robustness, which (probably, no ablation) show how also using state occupancy measures on top of the rest of the usual AIL machinery makes the agent transfer well to environment sharing the same state spaces, and differing only by dynamics. This observation directly echoes the main result of AIRL, which is that rewards robust to dynamics changes are only function of the state (not action). Discussing this result is a piece that is missing from the submission unfortunately.

The paper could use a good editing to address language errors.

---

> ### Author Response · Authors · 2020-11-20
> **Response to Official Review #4**
>
> Thank you for the thoughtful review. For a concise explanation we listed the main points by making quotation at each core of the review.
> * "Claims are neither properly defined and grounded in the literature, nor are they isolated experimentally": Our aim about the first claim of high variance was to generally proved this point empirically by comparison studies with three different AIL algorithms. Also, we believe that the transfer learning tasks verifies the second claim regarding domain shifts. The experiments of multi-goal environments deal with the third claim of termination biases. Compared to the standard continuous benchmarks, the multi-goal environments require the agent to take a considerably shorter time to end the episode (Table 2). For clarity, we have revised some expressions to show these points clearly, and we also provided ablation studies in the revision.
> * "The method addressing the (termination) bias seems to suffer from a logic flaw.": We believe there is a misunderstanding here, and we would like to argue this point. What Fig. 1 is displaying is not the actual rewards given by an environment. However, it shows two similar regularization methods for stochasticity of a policy, which have been widely applied in the RL domain when training stochastic policy. What we were trying to demonstrate in the figure is quite simple: 1) entropy-regularized RL/IRL methods had not been carefully considered for applying their regularization method after the ends of episodes, 2) the KL regularization method is one of the effortless ways to think about regularization after the termination, considering both actual reward $r(s_T,\cdot)$ and the regularization term $\mathrm{D}_\mathrm{KL}\bigl[\pi_E(\cdot|s_T)\big\Vert p_\mathrm{unif}(\cdot)\bigr]$ are thankfully zero.
> * "Clearer in the text that the method builds on AIRL" & "Discussing this result is a piece that is missing": Thank you for your suggestions. We agree with this point; we have used these critical points in our revision.
> * "It would also be interesting to evaluate the proposed IRL method on IRL tasks (reward recovery, as in AIRL)". We appreciate this suggestion. In Appendix E.5, we have plotted our reward function as a vector field, motivated by this comment and the work of Haarnoja et al. (2017).
> * "Disentangle the contribution of each component without ablation of the introduced techniques" & "proposed IRL method on IRL tasks (reward recovery, as in AIRL)": To answer these questions we included ablation studies and comparison results.

---

### Official Review · AnonReviewer2 · 2020-10-29
**Initial review for CAIRL**

**Rating:** 5
**Confidence:** 4

**Review:**

This paper builds on a recent inverse-RL method, AIRL. The authors argue that the rewards learned by AIRL are potentially inefficient since they depend on the ratio of state-action visitation distributions of the expert and the policy. To resolve this, CAIRL derives rewards that excludes these visitation distributions; this is realized in practice by employing another discriminator to approximate the state-visitation distribution ratio. The paper further proposes a mechanism to handle the reward-bias issue in IRL. Experiments with the MuJoCo locomotion tasks show that CAIRL is a competitive algorithm for imitation learning and can also handle dynamics mismatch between the expert and the learner.

I would like the authors to address the following concerns:

1.	Better motivation: It is argued that since AIRL rewards depend on the ratio of state-action visitations, it suffers due to the “high variance” of this ratio. Is there an analytic argument for this, or prior work to support the claim? Note that the density ratio is implicitly obtained with adversarial training, which has found empirical success across many ML domains. Moreover, CAIRL requires estimation of a similar ratio (of state-visitation densities), and this estimation feeds into the IRL reward computation. So why would CAIRL not be plagued with the same “high variance” issue?

2.	Reward-bias handling: The paper needs to provide more empirical evidence that the proposed reward-bias handling method is useful. I find the episode timesteps in Table 2 to be insufficient (especially with the variance in those numbers). The matter is made quite confusing by using a softplus on the IRL rewards – this manually enforces a positive bias on the IRL rewards. These are coupled with the negative rewards from the KL-penalty, and “alpha” now becomes a tunable knob to manage the overall reward-bias (as evident by “alpha” values in Table 3.). Note that DAC, which appears to be a much more principled way of handling the reward bias, does not add a modifier function (e.g. softplus) on the IRL rewards. I would encourage the authors to add more tasks where survival bias is detrimental (e.g. lunar-lander, etc.), and also comment on the role of softplus on their IRL rewards.

3.	Transfer-learning experiments: how was the algorithm (Algo 1.) changed for these experiments? The main section mentions the need for the transition-dynamics ratio in the IRL rewards. How do you obtain this ratio for these experiments?

4.	There is an L2-norm penalty added to the loss function (termed as REG). The paper says it is required to “make the algorithm well-defined”. Could you expand on this statement? Also, it would be interesting to do an ablation on this to measure its contribution to the overall performance of CAIRL. Connected question – are the baselines regularized with the REG and GP losses?

---

> ### Author Response · Authors · 2020-11-20
> **Response to Official Review #2**
>
> 1. **Better motivation.** We hypothesized that the logarithm of state-occupancy ratios, i.e., $\log\frac{\sum_{t=0}^\infty \gamma^t \Pr(s_t|\pi_E)}{\sum_{t=0}^\infty \gamma^t \Pr(s_t|\pi)}$, has been an unnecessary term to make AIL reward functions to be instructive rewards. Although it is not precisely an analytic argument, we have provided a helpful concept regarding state-conditioned reward in the revision (see Sec. 4.1 in the revision). We totally agree that non-linear approximators, such as neural networks, accomplished succeeding in numerous  tasks across many ML domains. However, we believe that what the neural networks can do is different from what a MaxEnt IRL reward function needs to represent. We stress that CAIRL requires estimation of the same ratio, but since we deliberately make use of the decoupled reward function $r_\theta$,  the estimation of the state-visitation densities does not directly feed into the actual reward computation.
> 2. **Reward-bias handling.** For these concerns, we have included ablation studies in our revision. One thing we would like to mention is that including absorbing in the DAC method requires the manipulation of adding synthetic transitions the both adversarial data and the corresponding terminal values; CAIRL does not change the trajectories and rewards using the KL regularization assumption in Table 2 and Sec. 4.3. As the terminal states always get the highest entropy bonuses (which is equal to 0 in the KL regularization), the positivity constraint does not exploit awareness of terminal states, but in some benchmarks, it has the practical advantage of preventing overly pessimistic rewards when the function is not sufficiently trained.
> 3. **Transfer learning.** Our reward formulation is summed up as follows.
>     * **In imitation learning:** MDPs of the expert and the agent are the same, especially, $\mathcal{P}(s^\prime | s, a)$ is identical for every possible cases. In this sense, the imitation learning setting is a special case of transfer learning that the CAIRL reward function learns the following quantity: $\gamma h_{\bar\psi} (s^\prime) +r_\theta(s,a) -  h_\psi(s) =  \log(\pi_E(a|s)/u)$.
>     * **In transfer learning:** If the dynamics of two MDPs are different, the CAIRL reward function will learn a more general form, i.e., $\gamma h_{\bar\psi} (s^\prime) +r_\theta(s,a) -  h_\psi(s) =  \log(\mathcal{P}_E(s^\prime|s,a)\cdot\pi_E(a|s)/\mathcal{P}(s^\prime|s,a)\cdot u)$.
>
>   Since the discriminator $D_{\theta,\psi}$is a function who takes $\mathcal{S}\times\mathcal{A}\times\mathcal{S}$ as the domain in the first place, the ratio $\log(\mathcal{P}_E(s^\prime|s,a)\cdot\pi_E(a|s)/\mathcal{P}(s^\prime|s,a)\cdot u)$ will be automatically obtained when the algorithm is applied in the situations when two agents reside in different MDPs. Therefore, one of our claims was: CAIRL can be applied to the standard Imitation learning problem and the specific transfer learning tasks that there are misalignments in dynamics. By unrolling the cumulative reward, we note that CAIRL minimizes the conditional distribution $\mathrm{D}_\mathrm{KL}\bigl[\Pr(a_0,s_1,\dots|s_0 = s,\pi)\Vert\Pr(a_0,s_1,\dots|s_0 = s,\pi_E)\bigr]$ (this last part is addressed in the revision).
>
> 4. **Regularization.** The expression "make the algorithm well-defined" was written in the purpose that there are two trainable parameters ($\theta$ and $\psi$) and only one logistic loss function: $\mathbb{E}_{\pi_E}\bigl[\log D(s,a,s^\prime)\bigr]+\mathbb{E}_\pi\bigl[\log (1 - D(s,a,s^\prime))\bigr]$, to train them; thus, a regularization method for one of the parameters is needed for convergence for the two parameters. Another point in the regularization method is that $\log\pi_E(a|s)$ and $\log(\pi_E(a|s)/u)$ are the optimally shaped reward functions for the entropy regularization and the KL regularization, respectively (Ng et al. 1999 & Sec. 4.2). In particular, we mention that $\bigl\lVert \gamma h_\bar\psi(s^\prime) - h_\psi (s) \bigr\rVert^2_2$ is a general choice for make reward functions to converge the optimal cases for both regularization methods (the only modification is to choose $\log\pi_\phi(a|s)$ and $\log(\pi_\phi(a|s)/u)$ in the $D_{\theta,\psi}$ architecture).

---

### Author Response · Authors · 2020-11-20
**Rebuttal Revision**

We much appreciate the three reviewers for their effort and fruitful reviews. To effectively answer the most critical questions, which cannot be readily answered by the official responses, we have decided to clarify our motivation and further formulations by revising the manuscript. Some of the concepts, especially regarding EBMs, were omitted in the original submission because we initially thought they might lead to too broad perspectives for one paper. However, we now believe the submission has been improved; it can relieve some critical concerns from the official reviews by discussing these concepts. Note that we also have included an ablation study (Table 2; the numbers are drawn now by a new experiments on five times larger tasks than the past version) and visualization results (Appendix E.5) in this version. Again, we thank the official reviewers for the positive influence that their reviews have provided.

---

### Decision · Program_Chairs · 2021-01-07
**Final Decision**

**Decision:**

Reject

**Comment:**

The paper was evaluated by 3 knowledgeable reviewers. All reviewers raised concerns about the motivation of the contribution of the paper. It is unclear why the use of an additional discriminator should reduce the variance of the log density ratio estimate. Also, the derivations were found to be not convincing or intuitive. These concerns have also not been alleviated after a rather extensive discussion of the reviewers with the authors. Moreover, the transfer setting to a new environment was unclear as it does not show how the  reward function transfers to new dynamics, so the transfer experiments rather evaluate how well the algorithm can imitate a policy on a different dynamics, but it does not tell that the extracted reward function is valid. While the experimental results seem promising, the authors are encouraged to improve the motivation of contribution, check which of the "incremental" contributions are very necessary and improve their evaluation on the transfer scenario.